# Latent Matters: Learning Deep State-Space Models

**Alexej Klushyn**[*][†][1]   **Richard Kurle**[1 4]   **Maximilian Soelch**[2]
**Botond Cseke**[2]   **Patrick van der Smagt**[2 3]

[1]Technical University of Munich   [2]Machine Learning Research Lab, Volkswagen Group, Munich
[3]Eötvös Loránd University   [4]AWS AI Labs

## Abstract

Deep state-space models (DSSMs) enable temporal predictions by learning the underlying dynamics of observed sequence data. They are often trained by maximising the evidence lower bound. However, as we show, this does not ensure the model actually learns the underlying dynamics. We therefore propose a constrained optimisation framework as a general approach for training DSSMs. Building upon this, we introduce the extended Kalman VAE (EKVAE), which combines amortised variational inference with classic Bayesian filtering/smoothing to model dynamics more accurately than RNN-based DSSMs. Our results show that the constrained optimisation framework significantly improves system identification and prediction accuracy on the example of established state-of-the-art DSSMs. The EKVAE outperforms previous models w.r.t. prediction accuracy, achieves remarkable results in identifying dynamical systems, and can furthermore successfully learn state-space representations where static and dynamic features are disentangled.

## 1   Introduction

Many dynamical systems can only be (partially) observed, with the exact dynamics unknown. Yet, precise models are needed for prediction and control [e.g. 19, 21, 25, 3]. Learning such accurate models is subject of current research, especially in image-based domains [e.g. 9].

Deep state-space models (DSSMs) [e.g. 28, 17, 13] describe sequence data by a (typically Markovian) nonlinear transition model and a nonlinear observation model. The transition model is assumed to capture the dynamics underlying the observed data, and the observation model maps the latent variables to the domain of observable data and accounts for the measurement noise. In this paper, we show that DSSMs, however, often do not learn the correct system dynamics, which is suboptimal for accurate predictions or performing downstream tasks, such as model-based reinforcement learning.

We identify in our experiments three main reasons causing this problem: (i) DSSMs are often trained by maximising the sequential evidence lower bound (ELBO). High ELBO values, however, do not imply the model has learned the correct system dynamics. (ii) The prior/initial distribution is usually just a Gaussian. This often leads to an over-regularisation of the approximate posterior or even to a broken generative model, where the transition model is not optimised to process samples from the prior. (iii) Most DSSMs use RNNs to approximate or support Bayesian filtering/smoothing. Yet, RNNs often prove to be a limiting factor for learning accurate models of the system dynamics; moreover, RNN-based transition models as in [8, 9] can lead to a non-Markovian state-space, where the latent variables do not capture the entire information about the system's state.

To address these problems, we propose the following solutions: (i) we introduce a constrained optimisation (CO) framework as a general approach for learning DSSMs. It ensures a good reconstruction

---

[*]Correspondence to `a.klushyn@tum.de`.   [†]Work done while at Machine Learning Research Lab, Volkswagen Group, Munich.

35th Conference on Neural Information Processing Systems (NeurIPS 2021).

quality and thus provides a necessary basis for learning the underlying system dynamics. To this end, we extend a recent method [16] presented in the context of variational autoencoders (VAEs) to DSSMs. We do this by formulating the sequential ELBO as the Lagrangian of a CO problem and introducing the associated optimisation algorithm. (ii) We complement the proposed CO framework with a powerful empirical Bayes prior. (iii) To obtain more accurate predictions of observed dynamical systems, we introduce the extended Kalman VAE (EKVAE), where we dispense with RNNs by combining extended Kalman filtering/smoothing with amortised variational inference and a neural linearisation approach. Furthermore, we show that the EKVAE is capable of learning state-space representations where static and dynamic features are disentangled. We use this to validate the learned model in the context of model-based reinforcement learning.

Our evaluation includes experiments on the image data of a moving pendulum [13] and on the reacher environment [26], where we use angle as well as high-dimensional RGB image data as observations. We show that each of our proposed approaches significantly helps in learning accurate models of observed dynamical systems—and that applying our CO framework to established DSSMs leads to a substantial increase in their prediction accuracy.

## 2  Background: A Rate–Distortion Perspective on Deep State-Space Models

DSSMs [e.g. 28, 17, 13] model an unknown distribution of observed sequence data $\mathbf{x}_{1:T} = (\mathbf{x}_1, \mathbf{x}_2, \ldots, \mathbf{x}_T)$ by means of typically lower-dimensional latent variables $\mathbf{z}_{1:T}$ that represent the underlying state of the system. To achieve this, the Markov assumption is imposed. It states that the future state $\mathbf{z}_{t+1}$ as well as the current observation $\mathbf{x}_t$ solely depend on $\mathbf{z}_t$:

$$p(\mathbf{x}_{1:T}, \mathbf{z}_{1:T} | \mathbf{u}_{1:T}) = p_\theta(\mathbf{x}_{1:T} | \mathbf{z}_{1:T}) \, p_\psi(\mathbf{z}_{1:T} | \mathbf{u}_{1:T})$$

$$= p(\mathbf{z}_1) \, p_\theta(\mathbf{x}_1 | \mathbf{z}_1) \prod_{t=2}^{T} p_\psi(\mathbf{z}_t | \mathbf{z}_{t-1}, \mathbf{u}_{t-1}) \, p_\theta(\mathbf{x}_t | \mathbf{z}_t), \tag{1}$$

where $\mathbf{u}_{1:T}$ are optional control signals (actions), and the use of different parameters $(\theta, \psi)$ will become important in the course of this paper. The model parameters in Eq. (1) are often learned through amortised variational inference [e.g. 17, 13]. This requires introducing a recognition model $q_\phi(\mathbf{z}_{1:T} | \mathbf{x}_{1:T}, \mathbf{u}_{1:T})$ that learns—in combination with the transition model $p_\psi(\mathbf{z}_t | \mathbf{z}_{t-1}, \mathbf{u}_{t-1})$—the dynamics underlying the observed data. The resulting objective function is known as sequential ELBO:

$$\log p(\mathbf{x}_{1:T} | \mathbf{u}_{1:T}) \geq \mathcal{F}_{\text{ELBO}}(\theta, \psi, \phi) = \mathbb{E}_{q_\phi} \left[ \log \frac{p_\theta(\mathbf{x}_{1:T} | \mathbf{z}_{1:T}) \, p_\psi(\mathbf{z}_{1:T} | \mathbf{u}_{1:T})}{q_\phi(\mathbf{z}_{1:T} | \mathbf{x}_{1:T}, \mathbf{u}_{1:T})} \right]. \tag{2}$$

In the context of generative models, the ELBO can be divided into a reconstruction term (distortion) and a compression term (rate) [1]. We extend the theory in [1] to DSSMs, where the distortion $\mathcal{D}(\theta, \phi) = - \mathbb{E}_{q_\phi} \left[ \log p_\theta(\mathbf{x}_{1:T} | \mathbf{z}_{1:T}) \right]$ optimises the model's ability for reconstructing observations, whereas the rate $\mathcal{R}(\phi, \psi) = \text{KL}\big(q_\phi(\mathbf{z}_{1:T} | \mathbf{x}_{1:T}, \mathbf{u}_{1:T}) \| p_\psi(\mathbf{z}_{1:T} | \mathbf{u}_{1:T})\big)$ enables learning the underlying dynamics. These definitions lead to the following general formulation of the ELBO:

$$\mathcal{F}_{\text{ELBO}}(\theta, \psi, \phi) = -\mathcal{D}(\theta, \phi) - \mathcal{R}(\psi, \phi). \tag{3}$$

Balancing the ratio between distortion and rate during optimisation can be an effective approach to improve the learning of DSSMs, as we discuss in the following section.

## 3  Constrained Optimisation Framework for Improved System Identification

High ELBO values do not necessarily imply that the model has learned the underlying system dynamics of the observed data, as we verify in Sec. 6.2. This is because different combinations of rate and distortion can result in the same ELBO value. Previous work addresses this issue by introducing weighting schedules for either $\mathcal{D}(\theta, \phi)$ or $\mathcal{R}(\psi, \phi)$ [e.g. 4] since a different ratio favours either better reconstruction or compression [1]. However, we demonstrate in Sec. 6 that balancing reconstruction and compression with predefined annealing schedules often does not achieve the desired result.

A more recent approach originates from the framework of VAEs: Rezende and Viola [23] and Klushyn et al. [16] define the VAE as a CO problem allowing for controlling the model's reconstruction quality.

We transfer this approach to DSSMs to ensure a good reconstruction—i.e. a low $\mathcal{D}(\theta, \phi)$—and thus provide a sufficient basis for learning the underlying system dynamics. To this end, we formulate the sequential ELBO as the Lagrangian of a CO problem (i) by specifying the rate $\mathcal{R}(\psi, \phi)$ in Eq. (3) as optimisation objective; and (ii) by imposing the inequality constraint $\mathcal{D}(\theta, \phi) \leq \mathcal{D}_0$. Here, $\mathcal{D}_0$ is a hyperparameter that defines the baseline for our desired reconstruction quality—we provide a heuristic for the simple determination of $\mathcal{D}_0$ in App. A.1. The resulting Lagrangian is

$$\mathcal{L}(\theta, \psi, \phi; \lambda) = \mathcal{R}(\psi, \phi) + \lambda\big(\mathcal{D}(\theta, \phi) - \mathcal{D}_0\big), \tag{4}$$

where the Lagrange multiplier $\lambda$ can be viewed as a weighting term for the distortion.

The original EM algorithm [e.g. 20] for optimising the ELBO, $\min_{\theta, \psi} \min_{\phi} -\mathcal{F}_{\text{ELBO}}(\theta, \psi, \phi)$, provides the following connection to the CO problem:

$$\overbrace{\min_{\theta, \psi}}^{\text{M-step}} \overbrace{\max_{\lambda} \min_{\phi}}^{\text{E-step}} \mathcal{L}(\theta, \psi, \phi; \lambda) \quad \text{s.t.} \quad \lambda \geq 0, \tag{5}$$

where, unlike in the original EM algorithm, we want $q_\phi(\mathbf{z}_{1:T} | \mathbf{x}_{1:T}, \mathbf{u}_{1:T})$ to additionally satisfy the inequality constraint $\mathcal{D}(\theta, \phi) \leq \mathcal{D}_0$ in the E-step. However, as detailed in [16], it can only be guaranteed that $\mathcal{L}(\theta, \psi, \phi; \lambda)$ optimises a lower bound on $\log p(\mathbf{x}_{1:T} | \mathbf{u}_{1:T})$ if and only if $1 \geq \lambda \geq 0$.

## 3.1 Learning the Initial Distribution

It is common practice to define the initial/prior distribution $p(\mathbf{z}_1)$ as a standard normal [e.g. 17, 8]. However, in this case, the prior KL in the ELBO can cause an over-regularisation (cf. [16]) of the approximate posterior and thus of the transition model. Furthermore, if the discrepancy between prior and posterior is too large, we may obtain a broken generative model, where $p(\mathbf{z}_t | \mathbf{z}_{t-1}, \mathbf{u}_{t-1})$ is not trained to process samples from $p(\mathbf{z}_1)$. We provide empirical evidence in Sec. 6.1.

This issue often arises in the context of neural models trained by stochastic gradient methods: in order to alleviate possible vanishing-gradient problems, time-series data is typically cut into equally-sized short-length units. For sufficiently large datasets, the initial states can therefore be assumed to cover most possible states, which results in a nontrivial marginal approximate posterior $\mathbb{E}_{p_\mathcal{D}}\big[q_\phi(\mathbf{z}_1 | \mathbf{x}, \mathbf{u})\big]$. Since the optimal prior distribution is $p^*(\mathbf{z}_1) = \mathbb{E}_{p_\mathcal{D}}\big[q_\phi(\mathbf{z}_1 | \mathbf{x}, \mathbf{u})\big]$ [cf. 27], an empirical Bayes prior $p_{\psi_0}(\mathbf{z}_1)$ must have the complexity to approximate $p_{\psi_0}(\mathbf{z}_1) \approx p^*(\mathbf{z}_1)$.

For this reason, we propose to learn a hierarchical prior $p_{\psi_0}(\mathbf{z}_1) = \int p_{\psi_0}(\mathbf{z}_1 | \boldsymbol{\zeta}) \, p(\boldsymbol{\zeta}) \, \mathrm{d}\boldsymbol{\zeta}$ as part of the DSSM by applying the variational approach in [16]. Klushyn et al. [16] define, by means of an approximate distribution $q_{\phi_0}(\boldsymbol{\zeta}|\mathbf{z}_1)$, a (VAE-like) lower bound on the optimal empirical Bayes prior:

$$\mathbb{E}_{p^*(\mathbf{z}_1)}\Big[\log p_{\psi_0}(\mathbf{z}_1)\Big] \geq \mathbb{E}_{p_\mathcal{D}(\mathbf{x}_{1:T}, \mathbf{u}_{1:T})} \mathbb{E}_{q_\phi(\mathbf{z}_{1:T} | \mathbf{x}_{1:T}, \mathbf{u}_{1:T})} \underbrace{\Big[\mathbb{E}_{q_{\phi_0}(\boldsymbol{\zeta}|\mathbf{z}_1)}\Big[\log \frac{p_{\psi_0}(\mathbf{z}_1 | \boldsymbol{\zeta}) \, p(\boldsymbol{\zeta})}{q_{\phi_0}(\boldsymbol{\zeta}|\mathbf{z}_1)}\Big]\Big]}_{= \mathcal{F}_{\text{VHP}}(\psi_0, \phi_0; \mathbf{z}_1)}, \tag{6}$$

where $p(\boldsymbol{\zeta})$ is a standard normal distribution and $p_\mathcal{D}(\mathbf{x}_{1:T}, \mathbf{u}_{1:T})$ is the empirical distribution of our data. This method is referred to as variational hierarchical prior (VHP). Learning the VHP as part of the model is consistent with the CO problem since Eq. (6) introduces an upper bound on the rate, $\mathcal{R}(\psi, \phi, \psi_0) \leq \mathcal{R}(\psi, \phi, \psi_0, \phi_0)$ (see App. A.2), and thus a lower bound on the ELBO. As a result, we obtain the Lagrangian $\mathcal{L}(\theta, \psi, \phi, \psi_0, \phi_0; \lambda) = \mathcal{R}(\psi, \phi, \psi_0, \phi_0) + \lambda\big(\mathcal{D}(\theta, \phi) - \mathcal{D}_0\big)$ and arrive at the following optimisation problem:

$$\overbrace{\min_{\psi_0, \phi_0}}^{\substack{\text{Empirical} \\ \text{Bayes}}} \overbrace{\min_{\theta, \psi}}^{\text{M-step}} \overbrace{\max_{\lambda} \min_{\phi}}^{\text{E-step}} \mathcal{L}(\theta, \psi, \phi, \psi_0, \phi_0; \lambda) \quad \text{s.t.} \quad \lambda \geq 0. \tag{7}$$

## 3.2 Optimisation Algorithm

In order to find the saddle point of the Lagrangian in Eq. (7), we propose Alg. 1, an extension of REWO [16] to DSSMs. Alg. 1 ensures that we optimise a lower bound on $\log p(\mathbf{x}_{1:T} | \mathbf{u}_{1:T})$ at the end of training, which is the case for $1 \geq \lambda \geq 0$. For this purpose, we apply a special update scheme

for $\lambda$, introduced and explained in depth in [16]:

$$\lambda^{(i)} = \lambda^{(i-1)} \cdot \exp\left[-\nu \cdot f_\lambda\left(\lambda^{(i-1)}, \mathcal{D}^{(i)}(\theta,\phi) - \mathcal{D}_0; \tau_1, \tau_2\right) \cdot \left(\mathcal{D}^{(i)}(\theta,\phi) - \mathcal{D}_0\right)\right]. \quad (8)$$

In this context, $i$ denotes the iteration step of the optimisation process. The function $f_\lambda$ is defined as $f_\lambda(\lambda, \delta; \tau_1, \tau_2) = \left(1 - H(\delta)\right) \cdot \tanh\left(\tau_1 \cdot \left(1/\lambda - 1\right)\right) - \tau_2 \cdot H(\delta)$, where $H$ is the Heaviside function, and $\tau_1$ as well as $\tau_2$ are slope parameters.

Furthermore, Alg. 1 allows us to efficiently learn the parameters of the VHP $(\psi_0, \phi_0)$ and the transition model $(\psi)$ by dividing the CO process into two phases: an initial and a main phase, which is the reason why we use different parameters for the transition and observation model. In the initial phase, the model is optimised w.r.t. $(\theta, \phi)$ to reduce the reconstruction error by learning the features of the individual observations $\mathbf{x}_t$. The main phase starts as soon as the inequality constraint $\mathcal{D}(\theta, \phi) \leq \mathcal{D}_0$ is satisfied. This serves as starting point for additionally optimising the parameters of the VHP $(\psi_0, \phi_0)$ and the transition model $(\psi)$, i.e. for learning the system dynamics.

---

**Algorithm 1** REWO for deep state-space models

---

Initialise $i = 1$
Initialise $\lambda^{(0)} = 1$
Initialise INITIALPHASE = TRUE
**while** training **do**
    Compute $\hat{\mathcal{D}}_{\text{ba}}$ (batch average)
    $\hat{\mathcal{D}}^{(i)} = (1 - \alpha) \cdot \hat{\mathcal{D}}_{\text{ba}} + \alpha \cdot \hat{\mathcal{D}}^{(i-1)}, \quad (\hat{\mathcal{D}}^{(0)} = \hat{\mathcal{D}}_{\text{ba}})$
    $\lambda^{(i)} \leftarrow \lambda^{(i-1)} \cdot \exp\left[-\nu \cdot f_\lambda\left(\lambda^{(i-1)}, \hat{\mathcal{D}}^{(i)} - \mathcal{D}_0; \tau_1, \tau_2\right) \cdot \left(\hat{\mathcal{D}}^{(i)} - \mathcal{D}_0\right)\right]$
    **if** $\hat{\mathcal{D}}^{(i)} \leq \mathcal{D}_0$ **then**
        INITIALPHASE = FALSE
    **end if**
    **if** INITIALPHASE **then**
        Optimise $\mathcal{L}(\theta, \psi, \phi, \psi_0, \phi_0; \lambda^{(i)})$ w.r.t. $\theta, \phi$
    **else**
        Optimise $\mathcal{L}(\theta, \psi, \phi, \psi_0, \phi_0; \lambda^{(i)})$ w.r.t. $\theta, \psi, \phi, \psi_0, \phi_0$
    **end if**
    $i \leftarrow i + 1$
**end while**

---

## 4 Extended Kalman VAE

The CO framework can be applied to any DSSM whose objective function is covered by the general rate–distortion formulation of the ELBO (Eq. (3)). We provide derivations for popular baseline models [17, 13] in App. A.3 and A.4. However, to achieve high prediction accuracies, the model itself should not prove to be a limiting factor for learning a precise description of the system dynamics.

Most DSSMs use deterministic RNNs as part of the recognition and/or transition model [e.g. 17, 13, 8, 9]. In [17], for instance, the parameters of the approximate posterior are learned through a (bidirectional) RNN, which is expected to replace classic Bayesian filtering/smoothing. The RNN-based transition model in [e.g. 8], on the other hand, allows combining amortised variational inference with Kalman filtering/smoothing [12, 22]. However, the use of RNNs often leads to less accurate models of the dynamical system, as we show in Sec. 6.2 and 6.3.

In order to increase the prediction accuracy, we introduce the extended Kalman VAE (EKVAE), where we dispense with RNNs by combining amortised variational inference with Bayesian filtering/smoothing. To compute the posterior, we leverage the concept of extended Kalman filters/smoothers [e.g. 11] but avoid the computationally expensive linearisation (Taylor expansion) of the transition and observation model. We achieve this (i) by directly learning the Jacobian of the dynamic model as a function of the current state, which we refer to as neural linearisation (Sec. 4.1); and (ii) by introducing a linear auxiliary-variable model similar to [8, 18] (Sec. 4.2). The EKVAE can be used as filter or smoother. In the following, we focus on the more complex **smoother version**—as it allows learning a more precise model [2]—and refer to App. A.6 for the filter version.

## 4.1 Neural Linearisation of the Dynamic Model

We model the nonlinear dynamical system by a Gaussian transition model that is locally linear w.r.t. discrete time steps [28, 13]:

$$p_\psi(\mathbf{z}_{t+1}|\,\mathbf{z}_t, \mathbf{u}_t) = \mathcal{N}\big(\mathbf{z}_{t+1}|\,\mathbf{F}_\psi(\mathbf{z}_t, \mathbf{u}_t)\,\mathbf{z}_t + \mathbf{B}_\psi(\mathbf{z}_t, \mathbf{u}_t)\,\mathbf{u}_t,\,\mathbf{Q}_\psi(\mathbf{z}_t, \mathbf{u}_t)\big), \tag{9}$$

where $\mathbf{F}_\psi$, $\mathbf{B}_\psi$, and $\mathbf{Q}_\psi$ are modelled by linear combinations of $M$ weighted base matrices:

$$\mathbf{F}_\psi(\mathbf{z}_t, \mathbf{u}_t) = \sum_{m=1}^{M} \alpha_\psi^{(m)}(\mathbf{z}_t, \mathbf{u}_t)\,\mathbf{F}^{(m)} \text{ etc., where } \boldsymbol{\alpha}_\psi(\mathbf{z}_t, \mathbf{u}_t) = \text{softmax}\big(\mathbf{g}_\psi(\mathbf{z}_t, \mathbf{u}_t)\big) \in \mathbb{R}^M. \tag{10}$$

The base matrices $\big\{\mathbf{F}^{(m)}, \mathbf{B}^{(m)}, \mathbf{Q}^{(m)}\big\}_{m=1}^{M}$ are learned parameters, and $\mathbf{g}_\psi(\mathbf{z}, \mathbf{u})$ is implemented as a neural network.

Next, we make the connection to extended Kalman filtering/smoothing, where the *prediction step* is based on the local Jacobian (first-order Taylor expansion) of the nonlinear transition function $\mathbf{z}_{t+1} = \mathbf{f}(\mathbf{z}_t, \mathbf{u}_t) + \mathbf{q}_t$, which is *unknown* in our case. Instead of computing the local Jacobian, however, our transition model (Eq. (9)) is designed to globally find the best linearisation at each time step as a function of the current state and action, which we refer to as neural linearisation.

This approach allows us to apply the extended Kalman filter or smoother algorithm, but replace the computationally expensive Taylor expansion in the *prediction step* with $\mathbf{F}_\psi(\mathbf{z}_t, \mathbf{u}_t)$ and $\mathbf{B}_\psi(\mathbf{z}_t, \mathbf{u}_t)$, as we derive in App. A.5 and verify in our experiments.

## 4.2 Linear Auxiliary-Variable Model

The observation model often needs to learn highly nonlinear functions, especially in case of high-dimensional sensory data, such as images. In order to enable an analytic computation of the posterior but avoid an expensive linearisation of the observation model, we introduce auxiliary variables $\mathbf{a}_{1:T}$ with a linear dependence on $\mathbf{z}_{1:T}$. As in [8, 18], we learn the nonlinear mapping from $\mathbf{a}_{1:T}$ to the high-dimensional observations $\mathbf{x}_{1:T}$ by a VAE's encoder–decoder pair, $q_\phi(\mathbf{a}_t|\,\mathbf{x}_t)$ and $p_\theta(\mathbf{x}_t|\,\mathbf{a}_t)$.

Since the dynamics are modelled by the transition function in $\mathbf{z}_{1:T}$—and $\mathbf{a}_t$ can be viewed as a low-dimensional representation of $\mathbf{x}_t$—we obtain the following observation model:

$$p(\mathbf{x}_{1:T}, \mathbf{a}_{1:T}|\,\mathbf{z}_{1:T}) = \prod_{t=1}^{T} p_\theta(\mathbf{x}_t|\,\mathbf{a}_t)\,p_\psi(\mathbf{a}_t|\,\mathbf{z}_t). \tag{11}$$

In contrast to [8, 18], we propose a time-invariant auxiliary-variable model,

$$p_\psi(\mathbf{a}_t|\,\mathbf{z}_t) = \mathcal{N}(\mathbf{a}_t|\,\mathbf{H}\,\mathbf{z}_t,\,\mathbf{R}), \tag{12}$$

where $\mathbf{H}$ and $\mathbf{R}$ are globally learned or predefined. The time-invariant $\mathbf{H}$ additionally allows us to learn disentangled state-space representations, as we elaborate at the end of this section.

By using $\mathbf{a}_{1:T}$ as pseudo observations, our *update step* corresponds to the classical Kalman filter/smoother algorithm because we do not need to linearise the observation model (see App. A.5). In combination with the neural linearisation approach, we can now analytically compute the filtered and smoothed distributions, $p_\psi(\mathbf{z}_t|\,\mathbf{a}_{1:t}, \mathbf{u}_{1:t-1})$ and $p_\psi(\mathbf{z}_t|\,\mathbf{a}_{1:T}, \mathbf{u}_{1:T-1})$, respectively. Note, however, that these are generally not optimal because we have a nonlinear Gaussian system that is locally linearised. As a result, we obtain the recognition model (**smoother version**):

$$q(\mathbf{z}_{1:T}, \mathbf{a}_{1:T}|\,\mathbf{x}_{1:T}, \mathbf{u}_{1:T}) = \prod_{t=1}^{T} p_\psi(\mathbf{z}_t|\,\mathbf{a}_{1:T}, \mathbf{u}_{1:T-1}) \prod_{t=1}^{T} q_\phi(\mathbf{a}_t|\,\mathbf{x}_t). \tag{13}$$

**Disentangling Static and Dynamic Features** In the context of latent-variable models, *disentanglement* typically means that different features are represented by different dimensions in latent space [e.g. 10]. DSSMs learn a representation of the system's state in the latent space. It can usually be split into *static* and *dynamic* features, e.g. the position and velocity of a robot arm, where the position can be inferred from a single frame, while the velocity requires a sequence of frames.

The EKVAE can be used to disentangle static and dynamic features due to its architecture: static features are represented separately by the auxiliary variables $\mathbf{a}_t$, which are learned via the encoder–decoder pair. By defining $\mathbf{H}$ in $p_\psi(\mathbf{a}_t|\mathbf{z}_t)$ (Eq. (12)) as rectangular identity matrix (cf. Fig. 5),

$$\mathbf{H} = (\delta_{ij}) \in \mathbb{R}^{D_\mathbf{a} \times D_\mathbf{z}}, \tag{14}$$

the model learns a latent representation where the first $D_\mathbf{a}$ dimensions of $\mathbf{z}_t$ correspond to static features $\mathbf{a}_t \in \mathbb{R}^{D_\mathbf{a}}$, such that $z_t^{(d)} = a_t^{(d)}$ for $d = 1, 2, \ldots, D_\mathbf{a}$. The remaining $D_\mathbf{z} - D_\mathbf{a}$ dimensions of $\mathbf{z}_t$ represent dynamic features, as we verify in Sec. 6.4.

### 4.3 Integration With the CO Framework: Deriving Distortion and Rate

To integrate the EKVAE with the CO framework introduced in Sec. 3, we define in the following the distortion $\mathcal{D}(\theta, \phi)$ and rate $\mathcal{R}(\psi, \phi, \psi_0, \phi_0)$ based on the transition, observation, and recognition model in Eqs. (9, 11, 13)—and the VHP in Eq. (6). A detailed derivation is provided in App. A.6.

The distortion is simply defined by the encoder–decoder pair:

$$\mathcal{D}(\theta, \phi) = -\sum_{t=1}^{T} \mathbb{E}_{q_\phi(\mathbf{a}_t|\mathbf{x}_t)} \Big[ \log p_\theta(\mathbf{x}_t|\mathbf{a}_t) \Big]. \tag{15}$$

Deriving the rate is more complicated: (i) we need to perform a sample-based optimisation of transition parameters $\psi$. This is especially crucial for $g_\psi(\mathbf{z}_t, \mathbf{u}_t)$ in Eq. (10), where an optimisation solely via extended Kalman smoothing, i.e. via deterministic mean values (cf. App. A.5), does not cover the range of application and would therefore result in a poorly trained transition model. (ii) Our recognition model $q(\mathbf{z}_{1:T}, \mathbf{a}_{1:T}|\mathbf{x}_{1:T}, \mathbf{u}_{1:T})$ (cf. Eq. (13)) does *not* contain the computationally more expensive pairwise smoothed distributions $p_\psi(\mathbf{z}_t, \mathbf{z}_{t-1}|\mathbf{a}_{1:T}, \mathbf{u}_{1:T-1})$ but only smoothed distributions $p_\psi(\mathbf{z}_t|\mathbf{a}_{1:T}, \mathbf{u}_{1:T-1})$. However, an optimisation of $p_\psi(\mathbf{z}_t|\mathbf{z}_{t-1}, \mathbf{u}_{t-1})$ based on samples $(\mathbf{z}_t, \mathbf{z}_{t-1})$ from smoothed distributions would lead to an inaccurate transition model.

To address these issues, we use the rate $\mathbb{E}_{q_\phi(\mathbf{a}_{1:T}|\mathbf{x}_{1:T})} \Big[ \sum_{t=1}^{T} \log \frac{q_\phi(\mathbf{a}_t|\mathbf{x}_t)}{p_\psi(\mathbf{a}_t|\mathbf{a}_{1:t-1}, \mathbf{u}_{1:t-1})} \Big]$ as our starting point. But instead of computing $p_\psi(\mathbf{a}_t|\mathbf{a}_{1:t-1}, \mathbf{u}_{1:t-1})$ analytically, we solve the corresponding integral (derived from the Bayesian filtering equations, see App. A.6.2) only w.r.t. $\mathbf{z}_t$ in closed form and marginalise $\mathbf{z}_{t-1}$ via Monte Carlo integration:

$$\log p_\psi(\mathbf{a}_t|\mathbf{a}_{1:t-1}, \mathbf{u}_{1:t-1}) = \log \iint p_\psi(\mathbf{a}_t|\mathbf{z}_t) p_\psi(\mathbf{z}_t|\mathbf{z}_{t-1}, \mathbf{u}_{t-1}) p_\psi(\mathbf{z}_{t-1}|\mathbf{a}_{1:t-1}, \mathbf{u}_{1:t-2}) \mathrm{d}\mathbf{z}_t \mathrm{d}\mathbf{z}_{t-1}$$

$$\geq \mathbb{E}_{p_\psi(\mathbf{z}_{t-1}|\mathbf{a}_{1:T}, \mathbf{u}_{1:T-1})} \left[ \log \frac{p_\psi(\mathbf{a}_t|\mathbf{z}_{t-1}, \mathbf{u}_{t-1}) \, p_\psi(\mathbf{z}_{t-1}|\mathbf{a}_{1:t-1}, \mathbf{u}_{1:t-2})}{p_\psi(\mathbf{z}_{t-1}|\mathbf{a}_{1:T}, \mathbf{u}_{1:T-1})} \right]. \tag{16}$$

In this context, the distribution $p_\psi(\mathbf{a}_t|\mathbf{z}_{t-1}, \mathbf{u}_{t-1})$ plays a crucial role as it includes all transition parameters $\psi$ and decouples $\mathbf{z}_t$ from $\mathbf{z}_{t-1}$. It therefore allows a sample-based optimisation of the transition model on the basis of the smoothed distribution $p_\psi(\mathbf{z}_{t-1}|\mathbf{a}_{1:T}, \mathbf{u}_{1:T-1})$. As a result—the complete derivation can be found in App. A.6.2—we obtain the following rate (**smoother version**):

$$\mathcal{R}(\psi, \phi, \psi_0, \phi_0) = \mathbb{E}_{q(\mathbf{z}_{1:T}, \mathbf{a}_{1:T}|\mathbf{x}_{1:T}, \mathbf{u}_{1:T})} \Big[ \mathcal{R}_{\text{initial}}(\psi, \phi, \psi_0, \phi_0; \mathbf{a}_{1:T}, \mathbf{z}_1)$$

$$+ \sum_{t=2}^{T} \left( \log \frac{q_\phi(\mathbf{a}_t|\mathbf{x}_t)}{p_\psi(\mathbf{a}_t|\mathbf{z}_{t-1}, \mathbf{u}_{t-1})} + \log \frac{p_\psi(\mathbf{z}_{t-1}|\mathbf{a}_{1:T}, \mathbf{u}_{1:T-1})}{p_\psi(\mathbf{z}_{t-1}|\mathbf{a}_{1:t-1}, \mathbf{u}_{1:t-2})} \right) \Big], \tag{17}$$

where the empirical Bayes prior introduced in Sec. 3.1 is learned via

$$\mathcal{R}_{\text{initial}}(\psi, \phi, \psi_0, \phi_0; \mathbf{a}_{1:T}, \mathbf{z}_1) = \log \frac{q_\phi(\mathbf{a}_1|\mathbf{x}_1)}{p_\psi(\mathbf{a}_1|\mathbf{z}_1)} + \log p_\psi(\mathbf{z}_1|\mathbf{a}_{1:T}, \mathbf{u}_{1:T-1}) - \mathcal{F}_{\text{VHP}}(\psi_0, \phi_0; \mathbf{z}_1). \tag{18}$$

Note that the log distributions in Eq. (17) and (18) can be expressed as closed-form KL divergences (see App. A.6.2). The distortion (Eq. (15)) and rate (Eq. (17)) now allow us to define the Lagrangian of the CO problem in Eq. (7) and thus to integrate the EKVAE with our CO framework.

# 5   Related Work

A popular method for avoiding local optima when maximising the ELBO is referred to as annealing [e.g. 4]. Here, the rate is multiplied by a weighting term $\beta$ that is increased from 0 to 1 during training. However, such predefined schedules often prove to be suboptimal, as we show in Tab. 1. For this reason, we extend the VAE-based approach in [16] to DSSMs by deriving a general Lagrangian formulation of the sequential ELBO on the basis of distortion and rate. This allows to represent the above weighting term by a Lagrange multiplier $\lambda = 1/\beta$, which is updated based on the reconstruction quality. Our proposed optimisation algorithm builds on [16] and includes several modifications to facilitate learning the underlying system dynamics, as we detail in Sec. 3.2.

Many DSSMs [e.g. 17, 8] use simple Gaussian prior/initial distributions, resulting in less accurate transition models (cf. Sec. 6.2 and App. A.7.2). In the empirical Bayes approach of [13], a separate recognition model learns an initial pseudo state $\boldsymbol{\zeta} \sim p(\boldsymbol{\zeta} | \mathbf{x}_{1:T}, \mathbf{u}_{1:T})$, which is then mapped to $\mathbf{z}_1 = f(\boldsymbol{\zeta})$ through a neural network. By contrast, the VAE-like empirical Bayes method (VHP) [16] that we use in our CO framework can directly substitute a Gaussian $p(\mathbf{z}_1)$ without further restrictions on the model architecture (cf. Sec. 3.1), which moreover leads to better results (see Tab. 1).

Since the introduction of stochastic gradient variational Bayes [15, 24], various extensions have been proposed for learning DSSMs via amortised variational inference, where, in contrast to the EKVAE (ours), classic Bayesian filtering/smoothing is approximated/replaced by deterministic RNNs [17, 13, 9, 5, 7, 29, 6]. Two popular examples—which we evaluate and integrate in our CO framework—are deep Kalman filters/smoothers (DKF/DKS) [17] and deep variational Bayes filters/smoothers (DVBF/DVBS) [13, 14]. Krishnan et al. [17] define two different recognition models based on uni-/bidirectional RNNs that parametrise the approximate filtered/smoothed distribution. In [13, 14], the approximate posterior is obtained by sharing parameters between the recognition and transition model; and an RNN is used for the initial time step, as described above. Although, DVBF/DVBS uses the same locally-linear transition model as the EKVAE, it does not take advantage of closed-form Bayesian inference, leading to a less accurate dynamic model, as we verify in Tab. 1.

Previous work has shown that hidden states of RNNs [8, 21] or probabilistic switch variables [18] can be used to predict the parameters of a (time-inhomogeneous) linear SSM in order to enable closed-form Bayesian inference. The Kalman VAE (KVAE) [8], for example, uses an auxiliary-variable model $p(\mathbf{x}_{1:T}, \mathbf{a}_{1:T}, \mathbf{z}_{1:T} | \mathbf{u}_{1:T}) = p(\mathbf{x}_{1:T} | \mathbf{a}_{1:T}) \, p(\mathbf{a}_{1:T} | \mathbf{z}_{1:T}) \, p(\mathbf{z}_{1:T} | \mathbf{u}_{1:T})$ and is based on linear Gaussian $p(\mathbf{a}_t | \mathbf{z}_t, \mathbf{h}_{t-1})$ and $p(\mathbf{z}_t | \mathbf{z}_{t-1}, \mathbf{h}_{t-1}, \mathbf{u}_{t-1})$, whose model parameters are conditioned on a deterministic hidden state $\mathbf{h}_{t-1} = \mathrm{LSTM}(\mathbf{a}_{1:t-1})$ for modelling nonlinear dynamics. This allows analytically computing the posterior by Kalman filtering/smoothing. In the EKVAE, we use a similar auxiliary-variable model as in [8], but dispense with RNNs/switch variables by choosing a transition model with a nonlinear dependence on $\mathbf{z}_{t-1}$. To compute the posterior, we leverage the concept of extended Kalman filtering/smoothing (cf. Sec. 4). This is beneficial as, for example, the LSTM-based deterministic path in the transition of [8] leads to a less accurate dynamic model and a non-Markovian state-space, meaning that not all information about the state is encoded in $\mathbf{z}_t$ (see Sec. 6.3).

# 6   Experiments

We validate our approach on image data of a moving pendulum and on the reacher environment of Deepmind's control suite [26], where we use angle as well as high-dimensional image data. The pendulum dataset was originally introduced in [13] and consists of 500 sequences with 15 images each, which have a size of $16 \times 16$ pixels. The reacher dataset consists of 2000 sequences with 30 time steps each. We use two versions in our experiments: (i) partially observed system states, i.e. the angles of the first and second joint; and (ii) RGB images of $64 \times 64$ pixels in size.

In our experiments, we use *smoothing* posteriors. This leads to more precise models [cf. 2] and allows inferring an accurate state-space representation of partially observed systems already in the initial time step. In Sec. 6.1, we demonstrate our CO framework on the example of deep Kalman smoothers (DKS, see App. A.3) [17]; and show in Sec. 6.2 that it significantly improves learning the system dynamics on the example of DKSs, deep variational Bayes smoothers (DVBSs, see App. A.4) [13, 14], and EKVAEs (ours). Furthermore, we verify in Sec. 6.3 that RNN-based transition models, as in KVAEs [8] and RSSMs [9], lead to a non-Markovian state-space. In Sec. 6.4, we show the benefits of disentangled state-space representations for model-based reinforcement learning.

## 6.1 Demonstrating the CO Framework on the Example of DKS

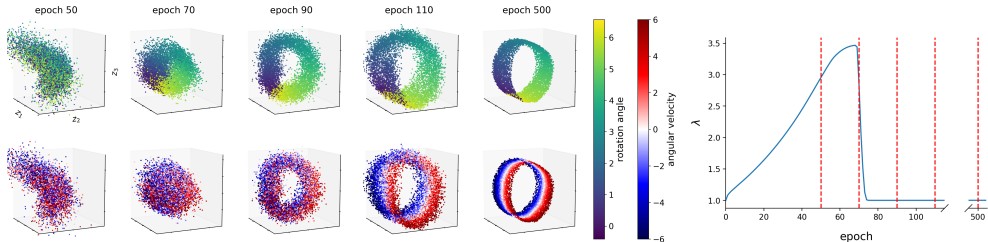

Figure 1: VHP-DKS (CO) on pendulum (image data). Learning the state-space representation with the CO framework. Distortion and rate are balanced by the Lagrange multiplier $\lambda$, which is updated (cf. Alg. 1) such that the model first improves the reconstruction quality/constraint by learning the rotation angle (see epoch 70). As soon as the constraint is satisfied, $\lambda$ decreases and the model starts learning the underlying dynamics, i.e. to represent the angular velocity. Robustness w.r.t. the hyperparameter $\mathcal{D}_0$ is demonstrated in App. A.7.1 (Fig. 9).

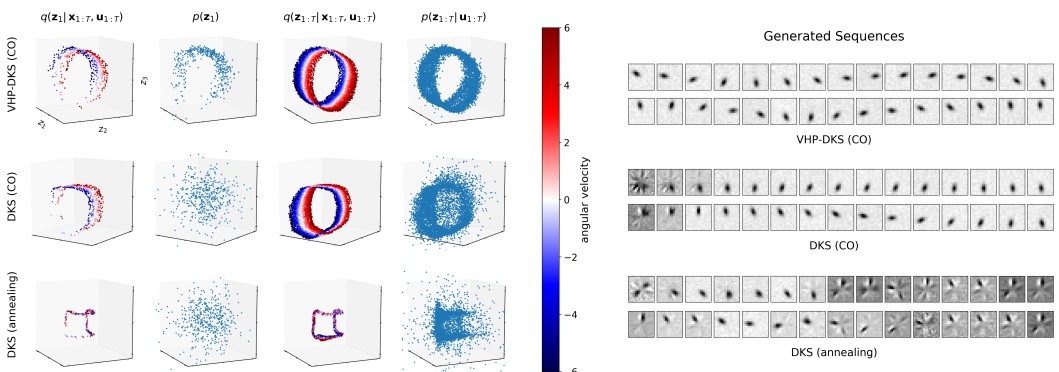

Figure 2: Pendulum (image data). In contrast to annealing (bottom), CO (middle & top) enables the model to learn the underlying dynamic system, as we verify in Table 1a. Furthermore, the VHP (top) significantly improves the quality of generated sequences. This is because the VHP learns a prior $p(\mathbf{z}_1) = \mathbb{E}_{p(\boldsymbol{\zeta})}\left[p(\mathbf{z}_1 \mid \boldsymbol{\zeta})\right]$ that matches the manifold of $\mathbb{E}_{p_\mathcal{D}}\left[q(\mathbf{z}_1 \mid \mathbf{x}_{1:T}, \mathbf{u}_{1:T})\right]$ (cf. first two columns of the latent-space visualisations).

Karl et al. [13] have shown that DKS is not capable of learning the angular velocity of the pendulum, i.e. to accurately predict the system, when trained classically or with annealing. In the following, we show that our CO framework solves this problem. We refer to the resulting model as VHP-DKS (CO), implying that it is trained via CO with the VHP as part of the model (see App. A.3 for the derivation).

Fig. 1 shows the optimisation process of VHP-DKS (CO): the model learns the underlying system dynamics, which is indicated by the barrel shape of the state-space representation [cf. 28, 13] and verified in Sec. 6.2. Complementary to this, we demonstrate in Fig. 2 that the VHP significantly improves the quality of generated sequences (no broken generative model) by learning a prior that matches $\mathbb{E}_{p_\mathcal{D}}\left[q(\mathbf{z}_1 \mid \mathbf{x}_{1:T}, \mathbf{u}_{1:T})\right]$—and show that annealing [4], in contrast to our CO framework, does not enable DKS to infer the angular velocity, which confirms the experimental findings in [13].

## 6.2 The Influence of the Learned State-Space Representation on the Prediction Accuracy

High ELBO values do not imply the model can accurately predict the observed system, as we show in Tab. 1. Our CO framework solves this problem: it improves system identification, leading to a significant increase in the prediction accuracy of the models (note the impact of the VHP). We evaluate if a system has been identified based on the correlation between inferred and ground-truth states—i.e. rotation angles and angular velocities—which is measured by $R^2$ of an OLS regression [cf. 13]. The prediction accuracy is evaluated by the MSE of 500 predicted sequences (pendulum/reacher: 15/30 times steps), conditioned on $\left\{\mathbf{z}_1^{(n)} \sim q(\mathbf{z}_1 \mid \mathbf{x}_{1:5}^{(n)}, \mathbf{u}_{1:4}^{(n)})\right\}_{n=1}^{500}$. This allows us to additionally verify the quality of the learned state-space representation in the initial time step. The EKVAE outperforms DKS and DVBS w.r.t. prediction accuracy and is even capable of identifying the dynamical system of reacher on the basis of $64 \times 64$ pixels RGB images (see Fig. 3 and Tab. 1c).

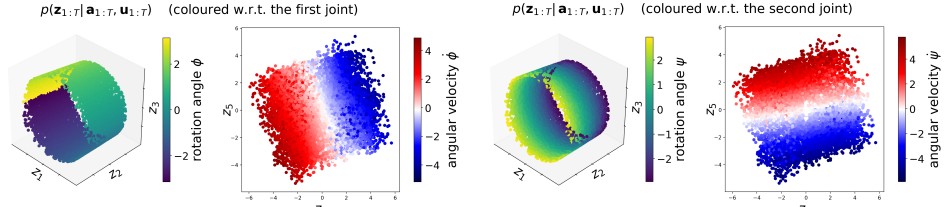

$p(\mathbf{z}_{1:T}|\mathbf{a}_{1:T}, \mathbf{u}_{1:T})$ (coloured w.r.t. the first joint)    $p(\mathbf{z}_{1:T}|\mathbf{a}_{1:T}, \mathbf{u}_{1:T})$ (coloured w.r.t. the second joint)

Figure 3: VHP-EKVAE (CO) on reacher (RGB image data). Five-dimensional state-space representation (disentangled): the first three dimensions ($z_1$, $z_2$, $z_3$) represent the two joint angles, the last two dimensions ($z_4$, $z_5$) represent the respective angular velocities. The barrel shape indicates the model has learned that the first joint can do a 360 degree turn; whereas the second joint is restricted to avoid self-collisions (cf. Fig. 7).

Table 1: The CO framework significantly improves system identification, as indicated by the high correlation ($R^2$ of an OLS regression) between inferred and ground-truth states. This leads to an increased prediction accuracy of the model, measured by the MSE of 500 predicted sequences conditioned on $\{\mathbf{z}_1^{(n)} \sim q(\mathbf{z}_1|\mathbf{x}_{1:5}^{(n)}, \mathbf{u}_{1:4}^{(n)})\}_{n=1}^{500}$. Note that high ELBO values do not imply the model can accurately predict the observed system.

(a) Pendulum (image data)

| model | test ELBO | $R^2_{\text{OLS reg.}}$ (ang. $\phi$) | $R^2_{\text{OLS reg.}}$ (vel. $\dot\phi$) | $\text{MSE}_{\text{predict}}^{\text{(smoothed)}}$ |
|---|---|---|---|---|
| VHP-EKVAE (CO) | **807.3** | **0.992** | **0.998** | **1.99E-4** |
| EKVAE (CO) | 805.9 | 0.957 | 0.991 | 3.53E-4 |
| EKVAE (annealing) | 804.2 | 0.687 | 0.339 | 1.94E-3 |
| VHP-DVBS (CO) | 804.3 | **0.992** | 0.989 | 5.63E-4 |
| DVBS (CO) | 803.8 | 0.985 | 0.980 | 9.41E-4 |
| DVBS (annealing) | 803.1 | 0.795 | 0.237 | 4.67E-3 |
| VHP-DKS (CO) | 804.7 | 0.973 | 0.990 | 1.73E-3 |
| DKS (CO) | 804.1 | 0.912 | 0.962 | 2.36E-3 |
| DKS (annealing) | 804.0 | 0.330 | 0.040 | 2.12E-2 |

**Bold** indicates the best result

Red indicates a low correlation with the ground truth

(b) Reacher (angle data)

| model | $R^2_{\text{OLS reg.}}$ (ang. $\phi$) | $R^2_{\text{OLS reg.}}$ (ang. $\psi$) | $R^2_{\text{OLS reg.}}$ (vel. $\dot\phi$) | $R^2_{\text{OLS reg.}}$ (vel. $\dot\psi$) | $\text{MSE}_{\text{predict}}^{\text{(smoothed)}}$ |
|---|---|---|---|---|---|
| VHP-EKVAE (CO) | 0.988 | **0.997** | 0.989 | 0.986 | **2.13E-5** |
| EKVAE (annealing) | 0.712 | 0.835 | 0.881 | 0.339 | 4.38E-4 |
| VHP-DVBS (CO) | **0.990** | 0.994 | 0.979 | **0.991** | 2.75E-4 |
| DVBS (annealing) | 0.897 | 0.949 | 0.963 | 0.778 | 4.17E-4 |
| VHP-DKS (CO) | 0.984 | 0.991 | 0.986 | 0.980 | 3.52E-4 |
| DKS (annealing) | 0.693 | 0.781 | 0.965 | 0.016 | 1.12E-3 |

(c) Reacher (RGB image data)

| model | $R^2_{\text{OLS reg.}}$ (ang. $\phi$) | $R^2_{\text{OLS reg.}}$ (ang. $\psi$) | $R^2_{\text{OLS reg.}}$ (vel. $\dot\phi$) | $R^2_{\text{OLS reg.}}$ (vel. $\dot\psi$) | $\text{MSE}_{\text{predict}}^{\text{(smoothed)}}$ |
|---|---|---|---|---|---|
| VHP-EKVAE (CO) | **0.980** | **0.986** | **0.991** | **0.987** | **1.64E-4** |
| EKVAE (annealing) | 0.672 | 0.052 | 0.668 | 0.091 | 1.82E-3 |

Supplementary to Tab. 1, we provide in App. A.7.2 (i) a statistic evaluation of different annealing schedules compared with CO, which is based on 25 runs each; (ii) visualisations of the state-space representations (initial time step *and* entire sequence) learned by the different models; and (iii) further evaluations including reconstructed, predicted, and generated sequences.

### 6.3 Comparison With the KVAE: Limitations of RNN-Based Transition Models

RNN-based transition models can lead to a non-Markovian state space, i.e. not all information about the state is encoded in $\mathbf{z}_t$, but partially in the RNN. This can significantly restrict (Bayesian) filtering and smoothing, resulting in a lower prediction accuracy of the model, as we show in Tab. 2 and Fig. 4.

The KVAE uses the transition model $p(\mathbf{z}_{t+1}|\mathbf{z}_t, \mathbf{h}_t, \mathbf{u}_t)$, where $\mathbf{h}_t = \text{LSTM}(\mathbf{a}_{1:t})$ (see Sec. 5). As shown in Tab. 2, this leads to a lower prediction accuracy compared to the EKVAE (cf. Tab. 1a); and predictions conditioned on the *smoothed* $\{\mathbf{z}_1 \sim p(\mathbf{z}_1|\mathbf{a}_{1:5}, \mathbf{u}_{1:4}), \mathbf{h}_1\}$ are significantly less accurate than predictions conditioned on the *filtered* $\{\mathbf{z}_5 \sim p(\mathbf{z}_5|\mathbf{a}_{1:5}, \mathbf{u}_{1:4}), \mathbf{h}_5\}$, which we explain in Fig. 4 and App. A.7.3. Note that the same applies to the RSSM, as we detail in App. A.7.3.

Table 2: Pendulum (image data), cf. Tab. 1a. As a consequence of the RNN-based transition models, the angular velocity is not encoded in $\mathbf{z}_t$ but in the RNN. This is indicated by the low correlation ($R^2$) with the ground-truth and verified by the low accuracy of the smoothing-based predictions, as we explain in Fig. 4 on the example of the KVAE.

| model | $R^2_{\text{OLS reg.}}$ (ang. $\phi$) | $R^2_{\text{OLS reg.}}$ (vel. $\dot\phi$) | $\text{MSE}_{\text{predict}}^{\text{(smoothed)}}$ | $\text{MSE}_{\text{predict}}^{\text{(filtered)}}$ |
|---|---|---|---|---|
| VHP-KVAE (CO) | 0.989 | 0.043 | 2.87E-3 | 4.24E-4 |
| KVAE (annealing) | 0.652 | 0.134 | 3.16E-3 | 6.67E-4 |
| VHP-RSSM (CO) | 0.915 | 0.086 | 3.10E-3 | 4.80E-4 |
| RSSM (annealing) | 0.158 | 0.060 | 3.23E-3 | 6.94E-4 |

Observations

Smoothing-Based Predictions Conditioned on $\{\mathbf{z}_1 \sim p(\mathbf{z}_1|\mathbf{a}_{1:5}, \mathbf{u}_{1:4}), \mathbf{h}_1 = \text{LSTM}(\mathbf{a}_1)\}$

Filtering-Based Predictions Conditioned on $\{\mathbf{z}_5 \sim p(\mathbf{z}_5|\mathbf{a}_{1:5}, \mathbf{u}_{1:4}), \mathbf{h}_5 = \text{LSTM}(\mathbf{a}_{1:5})\}$

Figure 4: VHP-KVAE (CO). The predictions show that the KVAE encodes the angular velocity of the pendulum in $\mathbf{h}_t = \text{LSTM}(\mathbf{a}_{1:t})$ and not in $\mathbf{z}_t$. This causes the poor smoothing-based predictions, as $\mathbf{h}_1 = \text{LSTM}(\mathbf{a}_1)$ does not have access to sequence data and therefore cannot infer the angular velocity.

## 6.4 Encoding Rewards: Policy Learning With Disentangled State-Space Representations

The EKVAE can learn state-space representations where static and dynamic features are disentangled (see Sec. 4.2). In the context of model-based reinforcement learning, these are often position and velocity, as we demonstrate on the example of pendulum (Fig. 5) and reacher (Fig. 3).

Such a disentangled representation allows us to use observations for encoding a goal position $\mathbf{p}_g = \mathbf{a}$ through $q(\mathbf{a}\,|\,\mathbf{x})$ or a goal velocity $\mathbf{v}_g$ through $p(\mathbf{z}\,|\,\mathbf{a}_{1:T}, \mathbf{u}_{1:T-1})\, q(\mathbf{a}_{1:T}\,|\,\mathbf{x}_{1:T})$, where $\mathbf{v}_g \in \mathbb{R}^{D_{\mathbf{v}}}$ is represented by the last $D_{\mathbf{v}} = D_{\mathbf{z}} - D_{\mathbf{a}}$ dimensions of $\mathbf{z}$ (cf. Sec. 4.2).

Therefore, if rewards are not available, the EKVAE can be used for defining reward functions based on an encoded $\mathbf{p}_g$ or $\mathbf{v}_g$, that target dimensions in $\mathbf{z}_t$ either representing the position or the velocity: $r_t^{\text{pos}}(\mathbf{z}_t, \mathbf{p}_g) = -\sum_{d=1}^{D_{\mathbf{a}}} \left(z_t^{(d)} - p_g^{(d)}\right)^2$ or $r_t^{\text{vel}}(\mathbf{z}_t, \mathbf{v}_g) = -\sum_{d=1}^{D_{\mathbf{v}}} \left(z_t^{(D_{\mathbf{a}}+d)} - v_g^{(d)}\right)^2$, where the negative mean squared error is a natural choice motivated by the Euclidean distance metric [cf. 16]. Depending on the task, we can use either $r_t^{\text{pos}}(\mathbf{z}_t, \mathbf{p}_g)$ or $r_t^{\text{vel}}(\mathbf{z}_t, \mathbf{v}_g)$ to learn a policy $\pi_\omega(\mathbf{u}_t\,|\,\mathbf{z}_t)$ by maximising $J(\omega) = \sum_{t=1}^{H-1} \mathbb{E}_{\pi_\omega(\mathbf{u}_t\,|\,\mathbf{z}_t)} \mathbb{E}_{p(\mathbf{z}_{t+1}\,|\,\mathbf{z}_t, \mathbf{u}_t)} \left[r_{t+1}\right]$, where $H$ is the planning horizon.

In Fig. 6 and 7, we use the above method to validate the EKVAE in the context of model-based reinforcement learning. Our results show that the EKVAE allows learning accurate policies without having access to (external) rewards. We provide experimental details and further results in App. A.7.4.

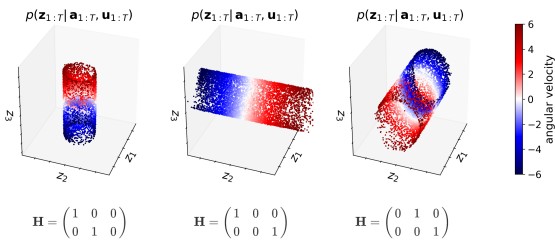

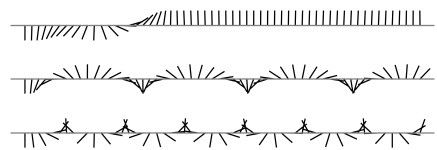

Figure 5: VHP-EKVAE (CO) on pendulum (image data). Disentangled (position–velocity) state-space representations: $\mathbf{H}$ defines the latent dimensions where the model learns to encode the rotation angle and the angular velocity.

Figure 6: VHP-EKVAE (CO). Visualisation of different policies that we learned based on the disentangled position–velocity representation in Fig. 5 (left): pendulum swing-up (top) and steady rotation (middle & bottom) by encoding the goal position for $r_t^{\text{pos}}(\mathbf{z}_t, \mathbf{p}_g)$ and the goal angular velocity for $r_t^{\text{vel}}(\mathbf{z}_t, \mathbf{v}_g)$, respectively.

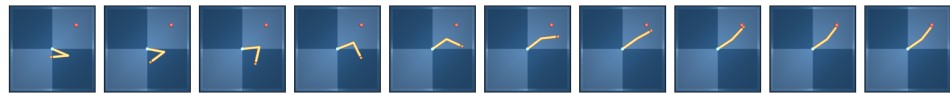

Figure 7: VHP-EKVAE (CO). Policy for reaching an encoded goal position (red dot) that we learned based on the disentangled (position–velocity) state-space representation in Fig. 3. This verifies that the EKVAE learns an accurate model of the reacher including self-collision avoidance.

## 7 Conclusion

In this paper, we have dealt with the question of how to learn DSSMs to obtain accurate predictions of observed dynamical systems. We have addressed the learning problem by proposing a CO framework for generic DSSMs. To this end, we have derived a general Lagrangian formulation of the sequential ELBO on the basis of distortion and rate—and extended the empirical Bayes prior (VHP) and the associated optimisation algorithm introduced in the context of VAEs to DSSMs. Building upon the CO framework, we have introduced the EKVAE, which combines extended Kalman filtering/smoothing with amortised variational inference and a neural linearisation approach.

Our experimental evaluations have demonstrated that applying the proposed CO framework to established DSSMs (e.g. DKF/DKS and DVBF/DVBS) facilitates system identification, with the VHP avoiding over-regularisation and broken generative models. The result is a substantial increase in prediction accuracy. In this context, we have shown that the EKVAE achieves a significantly higher prediction accuracy than state-of-the-art (RNN-based) models. Furthermore, we have shown that the EKVAE can learn disentangled position–velocity representations and demonstrated how these can be used for model-based reinforcement learning to define/encode reward functions and learn policies.

## Acknowledgements

We would like to thank Maximilian Karl and Djalel Benbouzid for valuable feedback and discussions.

## Funding Transparency Statement

None of the authors received related third party funding or third party support during the 36 months prior to the submission of this work. None of the authors had financial relationships with entities that could potentially be perceived to influence the submitted work in the 36 months prior to submission.

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
