# A Appendix

## A.1 Heuristic for Determining $\mathcal{D}_0$

In our experiments, we use the following heuristic for finding $\mathcal{D}_0$: first, the best distortion $\mathcal{D}_{\max}$ is determined, which the respective model achieves when trained via (classical) amortised variational inference; then the baseline for the desired reconstruction quality is defined as $\mathcal{D}_0 = 0.9 \, \mathcal{D}_{\max}$. Experimental support for this heuristic can be found in App. A.7.1 (Fig. 9).

## A.2 Learning the Initial Distribution

The VHP defines a (VAE-like) lower bound on the optimal empirical Bayes prior $p^*(\mathbf{z}_1)$:

$$\mathbb{E}_{p^*(\mathbf{z}_1)} \left[ \log p_{\psi_0}(\mathbf{z}_1) \right] = \mathbb{E}_{p_{\mathcal{D}}(\mathbf{x}, \mathbf{u})} \, \mathbb{E}_{q_\phi(\mathbf{z}_1 \mid \mathbf{x}, \mathbf{u})} \left[ \log p_{\psi_0}(\mathbf{z}_1) \right] \tag{19}$$

$$= \mathbb{E}_{p_{\mathcal{D}}(\mathbf{x}_{1:T}, \mathbf{u}_{1:T})} \, \mathbb{E}_{q_\phi(\mathbf{z}_{1:T} \mid \mathbf{x}_{1:T}, \mathbf{u}_{1:T})} \left[ \log p_{\psi_0}(\mathbf{z}_1) \right] \tag{20}$$

$$\geq \mathbb{E}_{p_{\mathcal{D}}(\mathbf{x}_{1:T}, \mathbf{u}_{1:T})} \, \mathbb{E}_{q_\phi(\mathbf{z}_{1:T} \mid \mathbf{x}_{1:T}, \mathbf{u}_{1:T})} \Big[ \underbrace{\mathbb{E}_{q_{\phi_0}(\boldsymbol{\zeta} \mid \mathbf{z}_1)} \left[ \log \frac{p_{\psi_0}(\mathbf{z}_1 \mid \boldsymbol{\zeta}) \, p(\boldsymbol{\zeta})}{q_{\phi_0}(\boldsymbol{\zeta} \mid \mathbf{z}_1)} \right]}_{= \, \mathcal{F}_{\text{VHP}}(\psi_0, \phi_0; \mathbf{z}_1)} \Big], \tag{21}$$

which introduces an upper bound on the rate:

$$\mathcal{R}(\psi, \phi, \psi_0) = \mathbb{E}_{q_\phi(\mathbf{z}_{1:T} \mid \mathbf{x}_{1:T}, \mathbf{u}_{1:T})} \left[ \log q_\phi(\mathbf{z}_{1:T} \mid \mathbf{x}_{1:T}, \mathbf{u}_{1:T}) - \log p_{\psi_0}(\mathbf{z}_1) - \sum_{t=2}^{T} \log p_\psi(\mathbf{z}_t \mid \mathbf{z}_{t-1}, \mathbf{u}_{t-1}) \right] \tag{22}$$

$$\leq \mathbb{E}_{q_\phi(\mathbf{z}_{1:T} \mid \mathbf{x}_{1:T}, \mathbf{u}_{1:T})} \left[ \log q_\phi(\mathbf{z}_{1:T} \mid \mathbf{x}_{1:T}, \mathbf{u}_{1:T}) - \mathcal{F}_{\text{VHP}}(\psi_0, \phi_0; \mathbf{z}_1) - \sum_{t=2}^{T} \log p_\psi(\mathbf{z}_t \mid \mathbf{z}_{t-1}, \mathbf{u}_{t-1}) \right] \tag{23}$$

$$= \mathcal{R}(\psi, \phi, \psi_0, \phi_0).$$

Note that $\mathcal{R}(\psi, \phi, \psi_0) \,\widehat{=}\, \mathcal{R}(\psi, \phi)$, where $\psi_0$ denotes the now learnable parameters of the prior. Hence, Eq. (23) leads to the following Lagrangian:

$$\mathcal{L}(\theta, \psi, \phi, \psi_0, \phi_0; \lambda) = \mathcal{R}(\psi, \phi, \psi_0, \phi_0) + \lambda \left( \mathcal{D}(\theta, \phi) - \mathcal{D}_0 \right), \tag{24}$$

with the corresponding constrained optimisation problem defined in Eq. (7).

### A.3 Integrating the Deep Kalman Filter and Smoother With the Constrained Optimisation Framework

In order to integrate DKF/DKS [17] with our proposed constrained optimisation framework, we specify the distortion and rate that define the ELBO. This allows us to formulate the Lagrangian of the constrained optimisation problem defined in Eq. (7).

#### A.3.1 Original Evidence Lower Bound (Smoother Version)

The objective function introduced by Krishnan et al. [17] for training deep Kalman smoothers (DKSs) is

$$\mathcal{F}_{\text{ELBO}}^{\text{DKS}}(\theta, \psi, \phi) = -\mathcal{D}_{\text{DKS}}(\theta, \phi) - \mathcal{R}_{\text{DKS}}(\psi, \phi). \tag{25}$$

The distortion is defined as

$$\mathcal{D}_{\text{DKS}}(\theta, \phi) = -\sum_{t=1}^{T} \mathbb{E}_{q_\phi(\mathbf{z}_t \mid \mathbf{x}_{1:T}, \mathbf{u}_{1:T})} \left[ \log p_\theta(\mathbf{x}_t \mid \mathbf{z}_t) \right], \tag{26}$$

and the rate is given by

$$\mathcal{R}_{\text{DKS}}(\psi, \phi) = \text{KL}\left( q_\phi(\mathbf{z}_1 \mid \mathbf{x}_{1:T}, \mathbf{u}_{1:T}) \| p(\mathbf{z}_1) \right)$$
$$+ \sum_{t=2}^{T} \mathbb{E}_{q_\phi(\mathbf{z}_{t-1} \mid \mathbf{x}_{1:T}, \mathbf{u}_{1:T})} \left[ \text{KL}\left( q_\phi(\mathbf{z}_t \mid \mathbf{z}_{t-1}, \mathbf{x}_{t:T}, \mathbf{u}_{t-1:T}) \| p_\psi(\mathbf{z}_t \mid \mathbf{z}_{t-1}, \mathbf{u}_{t-1}) \right) \right], \tag{27}$$

where $p(\mathbf{z}_1)$ is a standard normal distribution. See [17] for further implementation details. Note that the filter version (DKF) is obtained by replacing $q_\phi(\mathbf{z}_t \mid \mathbf{x}_{1:T}, \mathbf{u}_{1:T})$ with $q_\phi(\mathbf{z}_t \mid \mathbf{x}_{1:t}, \mathbf{u}_{1:t})$.

#### A.3.2 VHP-Based Evidence Lower Bound (Smoother Version)

In the following, we integrate the VHP with DKS:

$$\mathcal{F}_{\text{ELBO}}^{\text{VHP-DKS}}(\theta, \psi, \phi, \psi_0, \phi_0) = -\mathcal{D}_{\text{DKS}}(\theta, \phi) - \mathcal{R}_{\text{VHP-DKS}}(\psi, \phi, \psi_0, \phi_0). \tag{28}$$

The distortion remains identical to DKS (Eq. (26)). By replacing the prior $p(\mathbf{z}_1)$ in $\mathcal{R}_{\text{DKS}}(\psi, \phi)$ with the VHP defined in Eq. (6), we get:

$$\mathcal{R}_{\text{VHP-DKS}}(\psi, \phi, \psi_0, \phi_0) = \mathbb{E}_{q_\phi(\mathbf{z}_1 \mid \mathbf{x}_{1:T}, \mathbf{u}_{1:T})} \left[ \log q_\phi(\mathbf{z}_1 \mid \mathbf{x}_{1:T}, \mathbf{u}_{1:T}) - \mathcal{F}_{\text{VHP}}(\psi_0, \phi_0; \mathbf{z}_1) \right]$$
$$+ \sum_{t=2}^{T} \mathbb{E}_{q_\phi(\mathbf{z}_{t-1} \mid \mathbf{x}_{1:T}, \mathbf{u}_{1:T})} \left[ \text{KL}\left( q_\phi(\mathbf{z}_t \mid \mathbf{z}_{t-1}, \mathbf{x}_{t:T}, \mathbf{u}_{t-1:T}) \| p_\psi(\mathbf{z}_t \mid \mathbf{z}_{t-1}, \mathbf{u}_{t-1}) \right) \right]. \tag{29}$$

The filter version (VHP-DKF) is obtained, as with DKF, by replacing $q_\phi(\mathbf{z}_t \mid \mathbf{x}_{1:T}, \mathbf{u}_{1:T})$ with $q_\phi(\mathbf{z}_t \mid \mathbf{x}_{1:t}, \mathbf{u}_{1:t})$.

### A.4 Integrating the Deep Variational Bayes Filter and Smoother With the Constrained Optimisation Framework

In order to integrate DVBF/DVBS [13] with our proposed constrained optimisation framework, we specify the distortion and rate that define the ELBO. This allows us to formulate the Lagrangian of the constrained optimisation problem defined in Eq. (7).

#### A.4.1 Original Evidence Lower Bound (Smoother Version)

DVBF was originally introduced in [13]. In the following, we refer to the updated version presented in [14]. The locally-linear transition model is described in Sec. 4.1. The corresponding objective function for training deep variational Bayes smoothers (DVBSs) is

$$\mathcal{F}_{\text{ELBO}}^{\text{DVBS}}(\theta, \psi, \phi, \psi_0, \phi_0) = -\mathcal{D}_{\text{DVBS}}(\theta, \phi, \psi_0, \phi_0) - \mathcal{R}_{\text{DVBS}}(\psi, \phi, \psi_0, \phi_0). \tag{30}$$

The distortion is defined as

$$\mathcal{D}_{\text{DVBS}}(\theta, \phi, \psi_0, \phi_0) = - \mathbb{E}_{p_{\mathcal{D}}(\mathbf{x}_{1:T}, \mathbf{u}_{1:T})} \mathbb{E}_{q_{\phi_0}(\boldsymbol{\zeta} | \mathbf{x}_{1:T}, \mathbf{u}_{1:T})} \mathbb{E}_{q_\phi(\mathbf{z}_{2:T} | f_{\psi_0}(\boldsymbol{\zeta}), \mathbf{x}_{2:T}, \mathbf{u}_{1:T})} \left[ \log p_\theta(\mathbf{x}_1 | f_{\psi_0}(\boldsymbol{\zeta})) \right.$$

$$\left. + \sum_{t=2}^{T} \log p_\theta(\mathbf{x}_t | \mathbf{z}_t) \right], \tag{31}$$

where $f_{\psi_0}(\boldsymbol{\zeta}) \mathrel{\widehat{=}} \mathbf{z}_1$ mimics an empirical Bayes prior that is learned from data, and the approximate posterior distribution factorises as

$$q_\phi(\mathbf{z}_{2:T} | f_{\psi_0}(\boldsymbol{\zeta}), \mathbf{x}_{2:T}, \mathbf{u}_{1:T}) = q_\phi(\mathbf{z}_2 | f_{\psi_0}(\boldsymbol{\zeta}), \mathbf{x}_{2:T}, \mathbf{u}_{1:T}) \prod_{t=3}^{T} q_\phi(\mathbf{z}_t | \mathbf{z}_{t-1}, \mathbf{x}_{t:T}, \mathbf{u}_{t-1:T}). \tag{32}$$

Therefore, the rate is given by

$$\mathcal{R}_{\text{DVBS}}(\psi, \phi, \psi_0, \phi_0) = \mathbb{E}_{q_{\phi_0}(\boldsymbol{\zeta} | \mathbf{x}_{1:T}, \mathbf{u}_{1:T})} \mathbb{E}_{q_\phi(\mathbf{z}_{2:T} | f_{\psi_0}(\boldsymbol{\zeta}), \mathbf{x}_{2:T}, \mathbf{u}_{1:T})} \left[ \text{KL} \left( q_{\phi_0}(\boldsymbol{\zeta} | \mathbf{x}_{1:T}, \mathbf{u}_{1:T}) \| p(\boldsymbol{\zeta}) \right) \right.$$

$$+ \text{KL} \left( q_\phi(\mathbf{z}_2 | f_{\psi_0}(\boldsymbol{\zeta}), \mathbf{x}_{2:T}, \mathbf{u}_{1:T}) \| p_\psi(\mathbf{z}_2 | f_{\psi_0}(\boldsymbol{\zeta}), \mathbf{u}_1) \right)$$

$$\left. + \sum_{t=3}^{T} \text{KL} \left( q_\phi(\mathbf{z}_t | \mathbf{z}_{t-1}, \mathbf{x}_{t:T}, \mathbf{u}_{t-1:T}) \| p_\psi(\mathbf{z}_t | \mathbf{z}_{t-1}, \mathbf{u}_{t-1}) \right) \right], \tag{33}$$

where $p(\boldsymbol{\zeta})$ is a standard normal distribution. The conditional approximate posterior is implemented as the product of two distributions [14]:

$$q_\phi(\mathbf{z}_t | \mathbf{z}_{t-1}, \mathbf{x}_{t:T}, \mathbf{u}_{t-1:T}) \propto p_\phi(\mathbf{z}_t | \mathbf{z}_{t-1}, \mathbf{u}_{t-1}) \times q_\phi(\mathbf{z}_t | \mathbf{x}_{t:T}, \mathbf{u}_{t:T}). \tag{34}$$

Further implementation details can be found in [13] and [14]. Note that the filter version (DVBF) is obtained by replacing $q_\phi(\mathbf{z}_t | \mathbf{x}_{t:T}, \mathbf{u}_{t:T})$ in Eq. (34) with $q_\phi(\mathbf{z}_t | \mathbf{x}_t)$.

### A.4.2 VHP-Based Evidence Lower Bound (Smoother Version)

In the following, we integrate the VHP with DVBS:

$$\mathcal{F}_{\text{ELBO}}^{\text{VHP-DVBS}}(\theta, \psi, \phi, \psi_0, \phi_0) = -\mathcal{D}_{\text{VHP-DVBS}}(\theta, \phi) - \mathcal{R}_{\text{VHP-DVBS}}(\psi, \phi, \psi_0, \phi_0) \qquad (35)$$

By replacing the deterministic transformation $f_{\psi_0}(\boldsymbol{\zeta})$ with the VHP defined in Eq. (6), the marginal approximate posterior simplifies to $q_\phi(\mathbf{z}_t | \mathbf{x}_{t:T}, \mathbf{u}_{t:T})$ for all time steps *including the initial time step*. As a result, the approximate posterior factorises as

$$q_\phi(\mathbf{z}_{1:T} | \mathbf{x}_{1:T}, \mathbf{u}_{1:T}) = q_\phi(\mathbf{z}_1 | \mathbf{x}_{1:T}, \mathbf{u}_{1:T}) \prod_{t=2}^{T} q_\phi(\mathbf{z}_t | \mathbf{z}_{t-1}, \mathbf{x}_{t:T}, \mathbf{u}_{t-1:T}). \qquad (36)$$

Thus, the distortion is given by

$$\mathcal{D}_{\text{VHP-DVBS}}(\theta, \phi) = -\mathbb{E}_{q_\phi(\mathbf{z}_{1:T} | \mathbf{x}_{1:T}, \mathbf{u}_{1:T})} \left[ \sum_{t=1}^{T} \log p_\theta(\mathbf{x}_t | \mathbf{z}_t) \right], \qquad (37)$$

and the rate is defined as

$$\mathcal{R}_{\text{VHP-DVBS}}(\psi, \phi, \psi_0, \phi_0) = \mathbb{E}_{q_\phi(\mathbf{z}_{1:T} | \mathbf{x}_{1:T}, \mathbf{u}_{1:T})} \left[ \log q_\phi(\mathbf{z}_1 | \mathbf{x}_{1:T}, \mathbf{u}_{1:T}) - \mathcal{F}_{\text{VHP}}(\psi_0, \phi_0; \mathbf{z}_1) \right.$$
$$\left. - \sum_{t=2}^{T} \text{KL} \left( q_\phi(\mathbf{z}_t | \mathbf{z}_{t-1}, \mathbf{x}_{t:T}, \mathbf{u}_{t-1:T}) \| p_\psi(\mathbf{z}_t | \mathbf{z}_{t-1}, \mathbf{u}_{t-1}) \right) \right]. \qquad (38)$$

The conditional approximate posterior $q_\phi(\mathbf{z}_t | \mathbf{z}_{t-1}, \mathbf{x}_{t:T}, \mathbf{u}_{t-1:T})$ is implemented as for DVBS (Eq. (34)). The filter version (VHP-DVBF) is obtained, as with DVBF, by replacing $q_\phi(\mathbf{z}_t | \mathbf{x}_{t:T}, \mathbf{u}_{t:T})$ in Eq. (34) with $q_\phi(\mathbf{z}_t | \mathbf{x}_t)$.

## A.5 Extended Kalman Filtering and Smoothing With a Neural Linearisation of the Dynamic Model Function

In the following, we provide an analysis of how Kalman filtering/smoothing is applied in combination with the locally-linear transition model defined in Eq. (9) and the auxiliary-variable model defined in Eq. (12). To this end, we first consider the *prediction step* that allows analytically computing

$$p_\psi(\mathbf{z}_t \,|\, \mathbf{a}_{1:t-1}, \mathbf{u}_{1:t-1}) = \mathcal{N}(\mathbf{z}_t \,|\, \mathbf{m}_t^-, \mathbf{P}_t^-), \tag{39}$$

given the filtered distribution

$$p_\psi(\mathbf{z}_{t-1} \,|\, \mathbf{a}_{1:t-1}, \mathbf{u}_{1:t-1}) = \mathcal{N}(\mathbf{z}_{t-1} \,|\, \mathbf{m}_{t-1}, \mathbf{P}_{t-1}), \tag{40}$$

where $\mathbf{m}$ refers to the mean and $\mathbf{P}$ to the covariance of a Gaussian distribution.

The nonlinear dynamic model is typically defined as

$$\mathbf{z}_t = \mathbf{f}(\mathbf{z}_{t-1}, \mathbf{u}_{t-1}) + \mathbf{q}_{t-1}. \tag{41}$$

In extended Kalman filtering/smoothing, the dynamic model function $\mathbf{f}(\mathbf{z}, \mathbf{u})$ is locally linearised by means of a first-order Taylor expansion, which allows applying the Kalman filter/smoother algorithm as follows. The prediction step is defined by:

$$\mathbf{m}_t^- = \mathbf{f}(\mathbf{m}_{t-1}, \mathbf{u}_{t-1}) \tag{42}$$

$$\mathbf{F}_{t-1} = \left. \frac{\partial \mathbf{f}(\mathbf{z}, \mathbf{u})}{\partial \mathbf{z}} \right|_{\mathbf{m}_{t-1}, \mathbf{u}_{t-1}} \tag{43}$$

$$\mathbf{P}_t^- = \mathbf{F}_{t-1} \mathbf{P}_{t-1} \mathbf{F}_{t-1}^\mathsf{T} + \mathbf{Q}_{t-1}. \tag{44}$$

In case of an unknown dynamic model function, we can approximate $\mathbf{f}(\mathbf{z}, \mathbf{u})$ by a function that is locally linear w.r.t. discrete time steps. This allows formulating the prediction step of the mean as

$$\mathbf{m}_t^- = \left. \frac{\partial \mathbf{f}(\mathbf{z}, \mathbf{u})}{\partial \mathbf{z}} \right|_{\mathbf{m}_{t-1}, \mathbf{u}_{t-1}} \cdot \mathbf{m}_{t-1} + \left. \frac{\partial \mathbf{f}(\mathbf{z}, \mathbf{u})}{\partial \mathbf{u}} \right|_{\mathbf{m}_{t-1}, \mathbf{u}_{t-1}} \cdot \mathbf{u}_{t-1}. \tag{45}$$

In our proposed transition model (Eq. (9)), the above Jacobians are replaced by

$$\left. \frac{\partial \mathbf{f}(\mathbf{z}, \mathbf{u})}{\partial \mathbf{z}} \right|_{\mathbf{m}_{t-1}, \mathbf{u}_{t-1}} \rightarrow \mathbf{F}_\psi(\mathbf{m}_{t-1}, \mathbf{u}_{t-1}), \tag{46}$$

$$\left. \frac{\partial \mathbf{f}(\mathbf{z}, \mathbf{u})}{\partial \mathbf{u}} \right|_{\mathbf{m}_{t-1}, \mathbf{u}_{t-1}} \rightarrow \mathbf{B}_\psi(\mathbf{m}_{t-1}, \mathbf{u}_{t-1}). \tag{47}$$

Eq. (46) and (47) allow defining the prediction step as

$$\mathbf{m}_t^- = \mathbf{F}_\psi(\mathbf{m}_{t-1}, \mathbf{u}_{t-1}) \cdot \mathbf{m}_{t-1} + \mathbf{B}_\psi(\mathbf{m}_{t-1}, \mathbf{u}_{t-1}) \cdot \mathbf{u}_{t-1} \tag{48}$$

$$\mathbf{F}_{t-1} = \mathbf{F}_\psi(\mathbf{m}_{t-1}, \mathbf{u}_{t-1}) \tag{49}$$

$$\mathbf{Q}_{t-1} = \mathbf{Q}_\psi(\mathbf{m}_{t-1}, \mathbf{u}_{t-1}) \tag{50}$$

$$\mathbf{P}_t^- = \mathbf{F}_{t-1} \mathbf{P}_{t-1} \mathbf{F}_{t-1}^\mathsf{T} + \mathbf{Q}_{t-1}. \tag{51}$$

The *update step* corresponds to the classic Kalman filter/smoother due to the linear Gaussian $p_\psi(\mathbf{a}_t \,|\, \mathbf{z}_t)$ (Eq. (12)). The *backward recursion* is defined by $\mathbf{m}_t^-$, $\mathbf{F}_{t-1}$, and $\mathbf{P}_t^-$ in Eqs. (48, 49, 51). Therefore, it is identical to the Kalman smoother.

## A.6 Derivation of the Extended Kalman VAE

In the following, we derive $\mathcal{F}_{\text{ELBO}}$ of the EKVAE, i.e. the distortion $\mathcal{D}(\theta, \phi)$ and rate $\mathcal{R}(\psi, \phi, \psi_0, \phi_0)$ in Eq. (15) and (17). To this end, we start with the generative model that defines $p(\mathbf{x}_{1:T} | \mathbf{u}_{1:T})$. Note that the graphical model can be found in App. A.6.4.

### A.6.1 Generative Model

In addition to the latent variables $\mathbf{z}_t$, we use the auxiliary variables $\mathbf{a}_t$ to facilitate extended Kalman filtering/smoothing and $\boldsymbol{\zeta}$ to model the empirical Bayes prior:

$$p(\mathbf{x}_{1:T} | \mathbf{u}_{1:T})$$

$$= \iiint p(\mathbf{x}_{1:T}, \mathbf{a}_{1:T}, \mathbf{z}_{1:T}, \boldsymbol{\zeta} | \mathbf{u}_{1:T}) \, d\mathbf{a}_{1:T} \, d\mathbf{z}_{1:T} \, d\boldsymbol{\zeta} \tag{52}$$

$$= \iiint p(\mathbf{x}_{1:T} | \mathbf{a}_{1:T}, \cancel{\mathbf{z}_{1:T}}, \cancel{\boldsymbol{\zeta}}, \cancel{\mathbf{u}_{1:T-1}}) \, p(\mathbf{a}_{1:T} | \mathbf{z}_{1:T}, \cancel{\boldsymbol{\zeta}}, \cancel{\mathbf{u}_{1:T}}) \, p(\mathbf{z}_{1:T} | \boldsymbol{\zeta}, \mathbf{u}_{1:T}) \, p(\boldsymbol{\zeta}) \, d\mathbf{a}_{1:T} \, d\mathbf{z}_{1:T} \, d\boldsymbol{\zeta} \tag{53}$$

$$= \iiint p(\mathbf{x}_{1:T} | \mathbf{a}_{1:T}) \, p(\mathbf{a}_{1:T} | \mathbf{z}_{1:T}) \, p(\mathbf{z}_{1:T} | \boldsymbol{\zeta}, \mathbf{u}_{1:T}) \, p(\boldsymbol{\zeta}) \, d\mathbf{a}_{1:T} \, d\mathbf{z}_{1:T} \, d\boldsymbol{\zeta} \tag{54}$$

$$= \iiint \prod_{t=1}^{T} \Big( p_\theta(\mathbf{x}_t | \mathbf{a}_t) \, p_\psi(\mathbf{a}_t | \mathbf{z}_t) \Big) \prod_{t=2}^{T} \Big( p(\mathbf{z}_t | \mathbf{z}_{t-1}, \mathbf{u}_{t-1}) \Big) \, p(\mathbf{z}_1 | \boldsymbol{\zeta}) \, p(\boldsymbol{\zeta}) \, d\mathbf{a}_{1:T} \, d\mathbf{z}_{1:T} \, d\boldsymbol{\zeta}. \tag{55}$$

### A.6.2 Evidence Lower Bound (Smoother Version)

Starting from Eq. (55), in a first step, we marginalise $\mathbf{a}_{1:T}$ via Monte Carlo integration based on $q_\phi(\mathbf{a}_{1:T} | \mathbf{x}_{1:T})$. Furthermore, we use a chain factorisation based on Bayes' theorem to split the integral w.r.t. $\mathbf{z}_{1:T}$ into double integrals:

$$\log p(\mathbf{x}_{1:T} | \mathbf{u}_{1:T})$$

$$\geq \mathbb{E}_{q(\mathbf{a}_{1:T} | \mathbf{x}_{1:T})} \left[ \log \frac{p(\mathbf{x}_{1:T} | \mathbf{a}_{1:T})}{q(\mathbf{a}_{1:T} | \mathbf{x}_{1:T})} \right.$$
$$\left. + \log \iint \prod_{t=2}^{T} \Big( p(\mathbf{a}_t | \mathbf{z}_t) \, p(\mathbf{z}_t | \mathbf{z}_{t-1}, \mathbf{u}_{t-1}) \Big) \, p(\mathbf{a}_1 | \mathbf{z}_1) \, p(\mathbf{z}_1 | \boldsymbol{\zeta}) \, p(\boldsymbol{\zeta}) \, d\mathbf{z}_{1:T} \, d\boldsymbol{\zeta} \right] \tag{56}$$

$$= \mathbb{E}_{q(\mathbf{a}_{1:T} | \mathbf{x}_{1:T})} \left[ \log \frac{p(\mathbf{x}_{1:T} | \mathbf{a}_{1:T})}{q(\mathbf{a}_{1:T} | \mathbf{x}_{1:T})} + \log \iint p(\mathbf{a}_1 | \mathbf{z}_1) \, p(\mathbf{z}_1 | \boldsymbol{\zeta}) \, p(\boldsymbol{\zeta}) \, d\mathbf{z}_1 \, d\boldsymbol{\zeta} \right.$$
$$\left. + \sum_{t=2}^{T} \log \iint \underbrace{p(\mathbf{a}_t | \mathbf{z}_t)}_{\text{observation}} \overbrace{p(\mathbf{z}_t | \mathbf{z}_{t-1}, \mathbf{u}_{t-1})}^{\text{transition}} \underbrace{p(\mathbf{z}_{t-1} | \mathbf{a}_{1:t-1}, \mathbf{u}_{1:t-2})}_{\text{filtered distribution}} \, d\mathbf{z}_t \, d\mathbf{z}_{t-1} \right] \tag{57}$$

$$= \mathbb{E}_{q(\mathbf{a}_{1:T} | \mathbf{x}_{1:T})} \left[ \log \frac{p(\mathbf{x}_{1:T} | \mathbf{a}_{1:T})}{q(\mathbf{a}_{1:T} | \mathbf{x}_{1:T})} + \log \iint p(\mathbf{a}_1 | \mathbf{z}_1) \, p(\mathbf{z}_1 | \boldsymbol{\zeta}) \, p(\boldsymbol{\zeta}) \, d\mathbf{z}_1 \, d\boldsymbol{\zeta} \right.$$
$$\left. + \sum_{t=2}^{T} \log \int \overbrace{\underbrace{p(\mathbf{a}_t | \mathbf{z}_{t-1}, \mathbf{u}_{t-1})}_{\phantom{x}}}^{\int p(\mathbf{a}_t | \mathbf{z}_t) \, p(\mathbf{z}_t | \mathbf{z}_{t-1}, \mathbf{u}_{t-1}) \, d\mathbf{z}_t} \, p(\mathbf{z}_{t-1} | \mathbf{a}_{1:t-1}, \mathbf{u}_{1:t-2}) \, d\mathbf{z}_{t-1} \right] \tag{58}$$

As discussed in Sec. 4.3, we solve the integral in Eq. (58) only w.r.t. $\mathbf{z}_t$ in closed form and marginalise $\mathbf{z}_{t-1}$ in Eq. (59) via Monte Carlo integration based on the smoothed distributions $p_\psi(\mathbf{z}_{t-1}|\mathbf{a}_{1:T}, \mathbf{u}_{1:T-1})$. Furthermore, we marginalise $\zeta$ via Monte Carlo integration by using $\mathcal{F}_{\mathrm{VHP}}(\psi_0, \phi_0; \mathbf{z}_1)$ defined in Eq. (6):

$$(58) \geq \mathbb{E}_{q(\mathbf{a}_{1:T}|\mathbf{x}_{1:T})}\left[\log \frac{p(\mathbf{x}_{1:T}|\mathbf{a}_{1:T})}{q(\mathbf{a}_{1:T}|\mathbf{x}_{1:T})}\right.$$

$$+ \mathbb{E}_{p(\mathbf{z}_1|\mathbf{a}_{1:T},\mathbf{u}_{1:T-1})}\left[\log p(\mathbf{a}_1|\mathbf{z}_1) - \log p(\mathbf{z}_1|\mathbf{a}_{1:T}, \mathbf{u}_{1:T-1}) + \overbrace{\mathbb{E}_{q(\zeta|\mathbf{z}_1)}\left[\log \frac{p(\mathbf{z}_1|\zeta)\, p(\zeta)}{q(\zeta|\mathbf{z}_1)}\right]}^{\mathcal{F}_{\mathrm{VHP}}(\psi_0,\phi_0;\mathbf{z}_1)\text{ in Eq. (18)}}\right]$$

$$\left.+ \sum_{t=2}^{T}\mathbb{E}_{p(\mathbf{z}_{t-1}|\mathbf{a}_{1:T},\mathbf{u}_{1:T-1})}\left[\log p(\mathbf{a}_t|\mathbf{z}_{t-1},\mathbf{u}_{t-1}) + \log \frac{\overbrace{p(\mathbf{z}_{t-1}|\mathbf{a}_{1:t-1},\mathbf{u}_{1:t-2})}^{\text{filtered distribution}}}{\underbrace{p(\mathbf{z}_{t-1}|\mathbf{a}_{1:T},\mathbf{u}_{1:T-1})}_{\text{smoothed distribution}}}\right]\right] \qquad (59)$$

$$= \underbrace{\sum_{t=1}^{T}\mathbb{E}_{q(\mathbf{a}_t|\mathbf{x}_t)}\left[\log p(\mathbf{x}_t|\mathbf{a}_t)\right]}_{-\mathcal{D}(\theta,\phi)\text{ in Eq. (15)}}$$

$$- \mathbb{E}_{q(\mathbf{a}_{1:T}|\mathbf{x}_{1:T})}\left[\mathbb{E}_{p(\mathbf{z}_1|\mathbf{a}_{1:T},\mathbf{u}_{1:T-1})}\underbrace{\mathbb{E}_{q(\zeta|\mathbf{z}_1)}\left[\log \frac{q(\mathbf{a}_1|\mathbf{x}_1)}{p(\mathbf{a}_1|\mathbf{z}_1)} + \log \frac{p(\mathbf{z}_1|\mathbf{a}_{1:T},\mathbf{u}_{1:T-1})}{p(\mathbf{z}_1|\zeta)} + \log \frac{q(\zeta|\mathbf{z}_1)}{p(\zeta)}\right]}_{\mathcal{R}_{\text{initial}}(\psi,\phi,\psi_0,\phi_0;\mathbf{a}_{1:T},\mathbf{z}_1)\text{ in Eq. (17)}}\right.$$

$$\left.+ \sum_{t=2}^{T}\mathbb{E}_{p(\mathbf{z}_{t-1}|\mathbf{a}_{1:T},\mathbf{u}_{1:T-1})}\left[\log \frac{q(\mathbf{a}_t|\mathbf{x}_t)}{p(\mathbf{a}_t|\mathbf{z}_{t-1},\mathbf{u}_{t-1})} + \log \frac{p(\mathbf{z}_{t-1}|\mathbf{a}_{1:T},\mathbf{u}_{1:T-1})}{p(\mathbf{z}_{t-1}|\mathbf{a}_{1:t-1},\mathbf{u}_{1:t-2})}\right]\right] \qquad (60)$$

$$\approx \mathbb{E}_{\mathbf{z}_{1:T},\mathbf{a}_{1:T}\sim q(\mathbf{z}_{1:T},\mathbf{a}_{1:T}|\mathbf{x}_{1:T},\mathbf{u}_{1:T})}\mathbb{E}_{\zeta\sim q(\zeta|\mathbf{z}_1)}\left[\sum_{t=1}^{T}\log p(\mathbf{x}_t|\mathbf{a}_t)\right.$$

$$- \Big(\mathrm{KL}\big(q(\mathbf{a}_1|\mathbf{x}_1)\|\,p(\mathbf{a}_1|\mathbf{z}_1)\big) + \mathrm{KL}\big(p(\mathbf{z}_1|\mathbf{a}_{1:T},\mathbf{u}_{1:T-1})\|\,p(\mathbf{z}_1|\zeta)\big) + \mathrm{KL}\big(q(\zeta|\mathbf{z}_1)\|\,p(\zeta)\big)\Big)$$

$$\left.- \sum_{t=2}^{T}\Big(\mathrm{KL}\big(q(\mathbf{a}_t|\mathbf{x}_t)\|\,p(\mathbf{a}_t|\mathbf{z}_{t-1},\mathbf{u}_{t-1})\big) + \mathrm{KL}\big(p(\mathbf{z}_{t-1}|\mathbf{a}_{1:T},\mathbf{u}_{1:T-1})\|\,p(\mathbf{z}_{t-1}|\mathbf{a}_{1:t-1},\mathbf{u}_{1:t-2})\big)\Big)\right]$$

$$(61)$$

$$=: \mathcal{F}_{\mathrm{ELBO}}^{\mathrm{EKVAE\ (smoother\ version)}}$$

### A.6.3 Evidence Lower Bound (Filter Version)

The filter version of the EKVAE corresponds to the smoother version with the difference of replacing $p_\psi(\mathbf{z}_t \,|\, \mathbf{a}_{1:T}, \mathbf{u}_{1:T-1})$ by $p_\psi(\mathbf{z}_t \,|\, \mathbf{a}_{1:t}, \mathbf{u}_{1:t-1})$. In contrast to a closed-form evaluation, this enables a sample-based optimisation of the transition parameters $\psi$, as discussed in Sec. 4.3. As a result, we obtain:

$$
\log p(\mathbf{x}_{1:T} \,|\, \mathbf{u}_{1:T})
$$

$$
\geq \sum_{t=1}^{T} \mathbb{E}_{q(\mathbf{a}_t \,|\, \mathbf{x}_t)} \Big[ \log p(\mathbf{x}_t \,|\, \mathbf{a}_t) \Big]
$$

$$
- \mathbb{E}_{q(\mathbf{a}_{1:T} \,|\, \mathbf{x}_{1:T})} \Bigg[ \mathbb{E}_{p(\mathbf{z}_1 \,|\, \mathbf{a}_1)} \mathbb{E}_{q(\boldsymbol{\zeta} \,|\, \mathbf{z}_1)} \Bigg[ \log \frac{q(\mathbf{a}_1 \,|\, \mathbf{x}_1)}{p(\mathbf{a}_1 \,|\, \mathbf{z}_1)} + \log \frac{p(\mathbf{z}_1 \,|\, \mathbf{a}_1)}{p(\mathbf{z}_1 \,|\, \boldsymbol{\zeta})} + \log \frac{q(\boldsymbol{\zeta} \,|\, \mathbf{z}_1)}{p(\boldsymbol{\zeta})} \Bigg]
$$

$$
+ \sum_{t=2}^{T} \mathbb{E}_{p(\mathbf{z}_{t-1} \,|\, \mathbf{a}_{1:t-1}, \mathbf{u}_{1:t-2})} \Bigg[ \log \frac{q(\mathbf{a}_t \,|\, \mathbf{x}_t)}{p(\mathbf{a}_t \,|\, \mathbf{z}_{t-1}, \mathbf{u}_{t-1})} \Bigg] \Bigg] \tag{62}
$$

$$
\approx \mathbb{E}_{\mathbf{z}_{1:T}, \mathbf{a}_{1:T} \sim q(\mathbf{z}_{1:T}, \mathbf{a}_{1:T} \,|\, \mathbf{x}_{1:T}, \mathbf{u}_{1:T})} \mathbb{E}_{\boldsymbol{\zeta} \sim q(\boldsymbol{\zeta} \,|\, \mathbf{z}_1)} \Bigg[ \sum_{t=1}^{T} \log p(\mathbf{x}_t \,|\, \mathbf{a}_t)
$$

$$
- \Big( \mathrm{KL}\big( q(\mathbf{a}_1 \,|\, \mathbf{x}_1) \,\|\, p(\mathbf{a}_1 \,|\, \mathbf{z}_1) \big) + \mathrm{KL}\big( p(\mathbf{z}_1 \,|\, \mathbf{a}_1) \,\|\, p(\mathbf{z}_1 \,|\, \boldsymbol{\zeta}) \big) + \mathrm{KL}\big( q(\boldsymbol{\zeta} \,|\, \mathbf{z}_1) \,\|\, p(\boldsymbol{\zeta}) \big) \Big)
$$

$$
- \sum_{t=2}^{T} \mathrm{KL}\big( q(\mathbf{a}_t \,|\, \mathbf{x}_t) \,\|\, p(\mathbf{a}_t \,|\, \mathbf{z}_{t-1}, \mathbf{u}_{t-1}) \big) \Bigg] \tag{63}
$$

$$
=: \mathcal{F}_{\mathrm{ELBO}}^{\mathrm{EKVAE\ (filter\ version)}}
$$

### A.6.4 Graphical Model

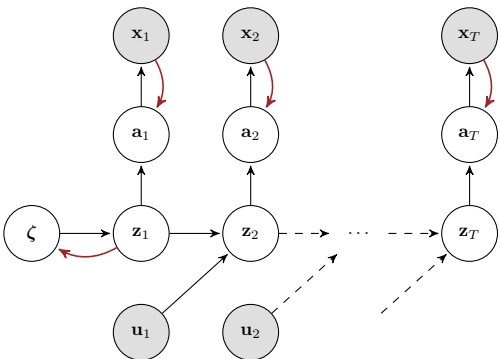

Figure 8: Graphical model of the EKVAE. Red arrows indicate the variational inference networks used for the auxiliary variables $\mathbf{a}_t$ and the VHP variable $\boldsymbol{\zeta}$. Given samples $\mathbf{a}_{1:T}$, the states $\mathbf{z}_{1:T}$ are inferred through (closed-form) Bayesian filtering/smoothing using the locally-linear transition model in Eq. (9) and the time-invariant auxiliary-variable model in Eq. (12).

## A.7 Supplementary Experimental Results

### A.7.1 Demonstrating the CO Framework on the Example of DKS

The robustness of the CO framework w.r.t. the hyperparameter $\mathcal{D}_0$ is demonstrated in Fig. 9. For this purpose, we evaluate the correlation of the inferred with the ground-truth angular velocity as a function of $\mathcal{D}_0$—all other hyperparamters are kept constant. The evaluations are based on 25 runs each using a random seed.

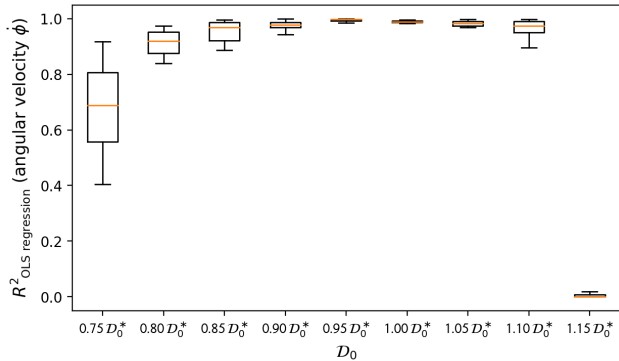

Figure 9: Pendulum (image data). To evaluate the robustness of the CO framework w.r.t. $\mathcal{D}_0$, we measure the correlation between inferred and ground-truth angular velocity by $R^2$ (OLS regression). Here, $\mathcal{D}_0^*$ corresponds to the value determined by our heuristic (see App. A.1). Note that the abrupt drop at $1.15\,\mathcal{D}_0^*$ is due to the fact that the constraint cannot be satisfied (cf. heuristic), leading to an ill-posed constrained optimisation problem and thus to a poorly trained model.

### A.7.2 The Influence of the Learned State-Space Representation on the Prediction Accuracy

The pendulum can do a full 360 degree turn; therefore, the models in Tab. 1a learn to represent the rotation angle $\phi$ by a circle, resulting in a barrel-shaped state-space representation (cf. Fig. 1). To this end, we perform three OLS regressions on the learned representations. In the first two, we use $\sin(\phi)$ and $\cos(\phi)$ as ground truth [cf. 13], where $R^{2\ (\text{ang. }\phi)}_{\text{OLS reg.}}$ is the corresponding mean. $R^{2\ (\text{vel. }\dot{\phi})}_{\text{OLS reg.}}$ refers to the third OLS regression with $\dot{\phi}$ as ground truth.

As in the pendulum experiments, we measure the correlation between inferred and ground-truth states through $R^2$ of an OLS regression. In case of angle data (Tab. 1b), we perform *four* OLS regressions on the learned representations with $(\phi, \psi, \dot{\phi}, \dot{\psi})$ as ground truth. In case of image data (Tab. 1c), we perform *five* OLS regressions on the learned representations because we use $\sin(\phi)$ and $\cos(\phi)$, instead of $\phi$, as ground truth, where $R^{2\ (\text{ang. }\phi)}_{\text{OLS reg.}}$ refers to the corresponding mean. Similar to the pendulum, this is necessary for image data since the model learns to represent the first joint angle $\phi$ of the reacher by a circle (cf. Fig. 3). This is because the first joint can do, in contrast to the second one, a full 360 degree turn.

In the following, we provide: (i) a statistic evaluation of different annealing schedules compared with CO, which is based on 25 runs each using a random seed (see Fig. 10); (ii) visualisations of the state-space representations learned by the different models (see Figs. 11–19); (iii) further evaluations including reconstructed, predicted, and generated sequences (see Figs. 11–21).

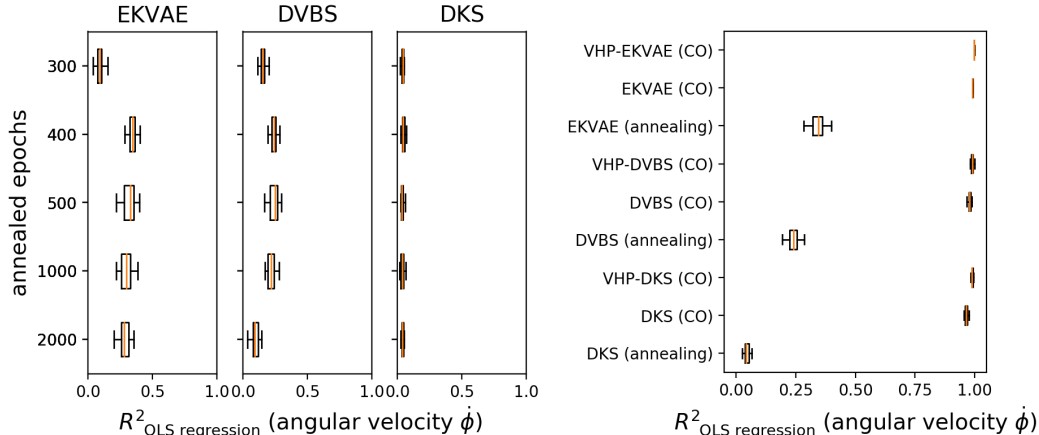

Figure 10: Pendulum (image data). Statistic evaluation of different annealing schedules. For this purpose, we measure the correlation between inferred and ground-truth angular velocity by $R^2$ (OLS regression) and compare the best schedule with CO (right), cf. Tab. 1a in Sec. 6.2. The statistics are based on 25 experimental runs each using a random seed and indicate that CO facilitates system identification.

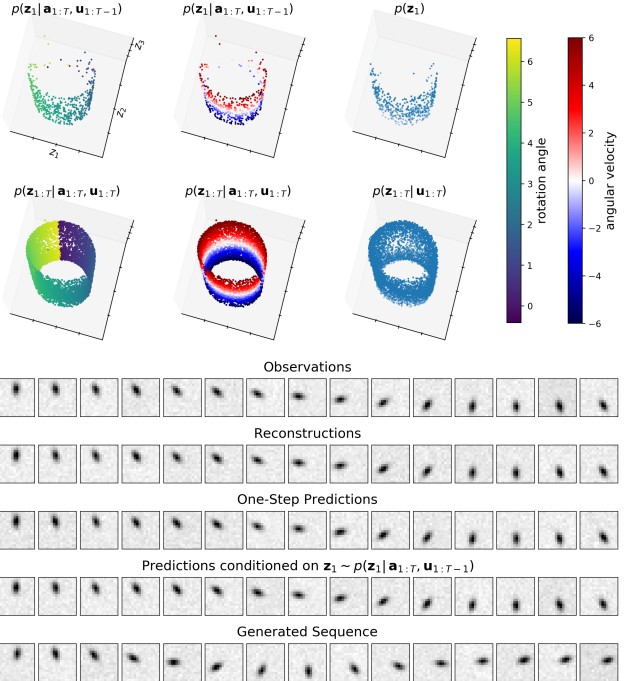

Figure 11: VHP-EKVAE (CO) trained on pendulum image data (supplementary to Tab. 1a in Sec. 6.2). In combination with the constrained optimisation framework, the EKVAE identifies the dynamical system of the pendulum and learns to predict it accurately.

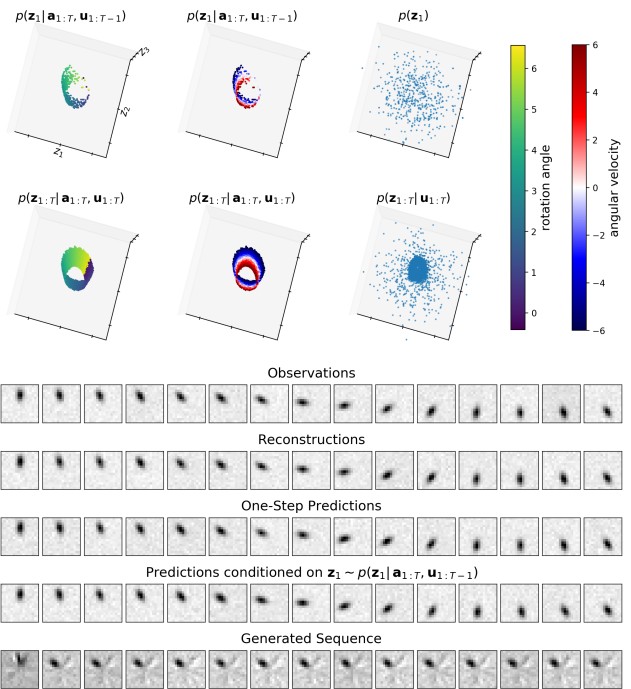

Figure 12: EKVAE (CO) trained on pendulum image data (supplementary to Tab. 1a in Sec. 6.2). Without the VHP, the EKVAE identifies the dynamical system of the pendulum but does not learn to process samples from the prior, which results in a broken generative model.

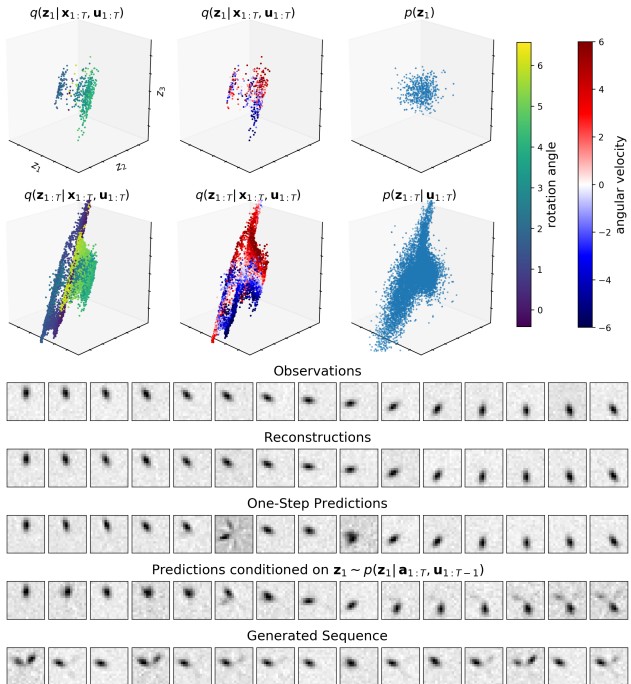

Figure 13: EKVAE (annealing) trained on pendulum image data (supplementary to Tab. 1a in Sec. 6.2). Without the constrained optimisation framework, the EKVAE does not learn to accurately predict the observed dynamical system.

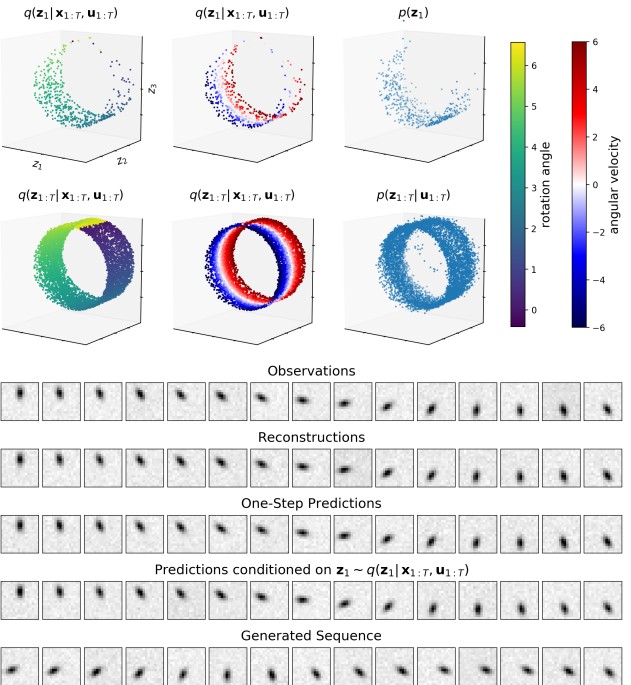

Figure 14: VHP-DVBS (CO) trained on pendulum image data (supplementary to Tab. 1a in Sec. 6.2). In combination with the constrained optimisation framework, DVBS identifies the dynamical system of the pendulum and learns to predict it accurately.

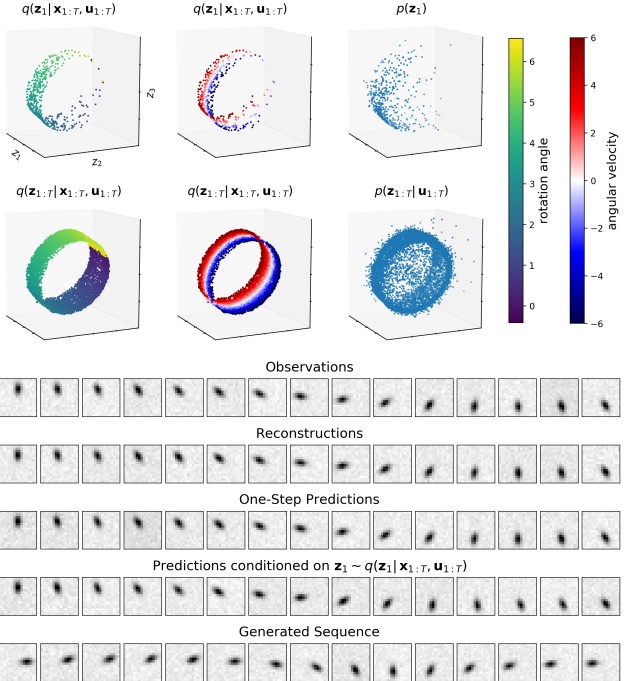

Figure 15: DVBS (CO) trained on pendulum image data (supplementary to Tab. 1a in Sec. 6.2). The original empirical Bayes prior proposed for DVBS leads to a poorer generative model than the VHP.

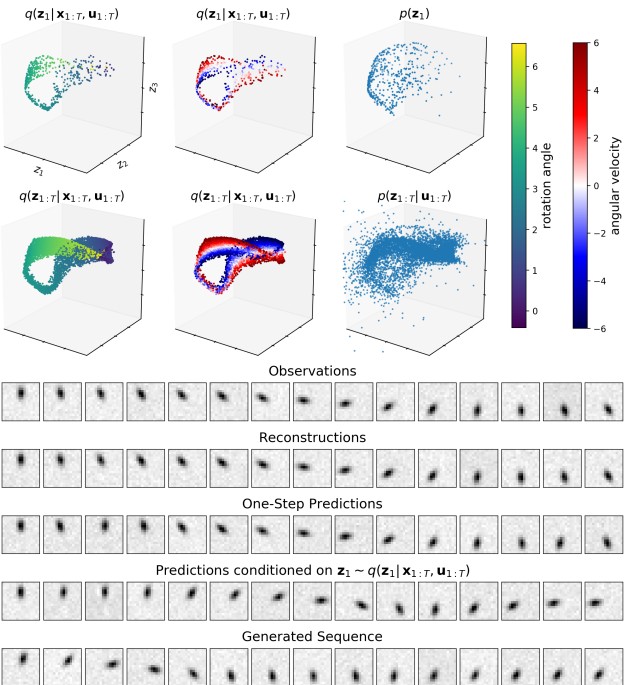

Figure 16: DVBS (annealing) trained on pendulum image data (supplementary to Tab. 1a in Sec. 6.2). Without the constrained optimisation framework, DVBS does not learn to accurately predict the observed dynamical system.

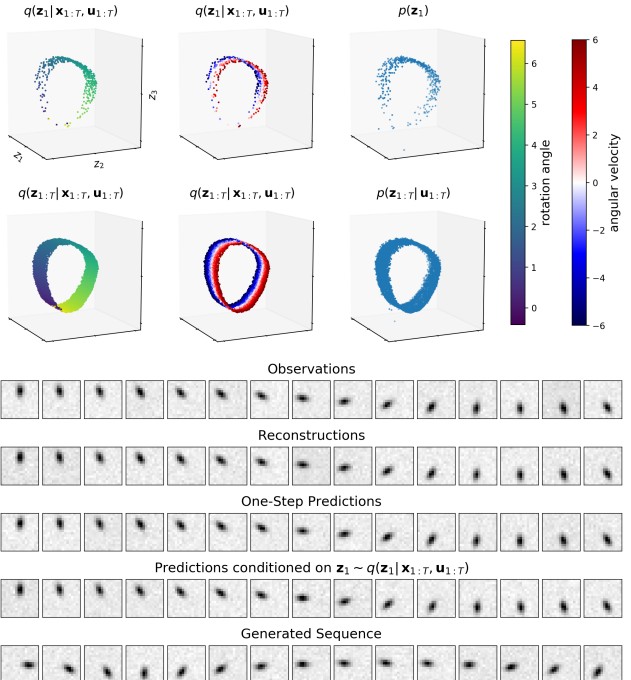

Figure 17: VHP-DKS (CO) trained on pendulum image data (supplementary to Tab. 1a in Sec. 6.2). In combination with the constrained optimisation framework, DKS identifies the dynamical system of the pendulum and learns to predict it accurately.

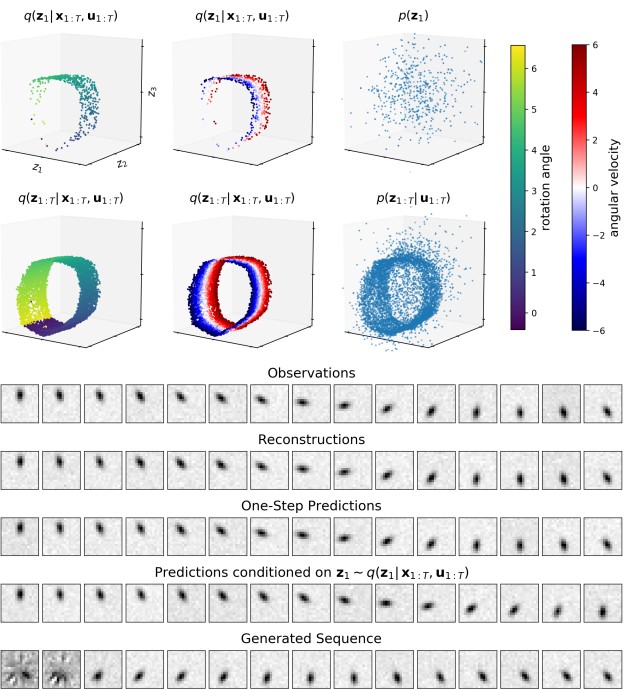

Figure 18: DKS (CO) trained on pendulum image data (supplementary to Tab. 1a in Sec. 6.2). Without the VHP, DKS identifies the dynamical system of the pendulum but does not learn to process samples from the prior, which results in a broken generative model.

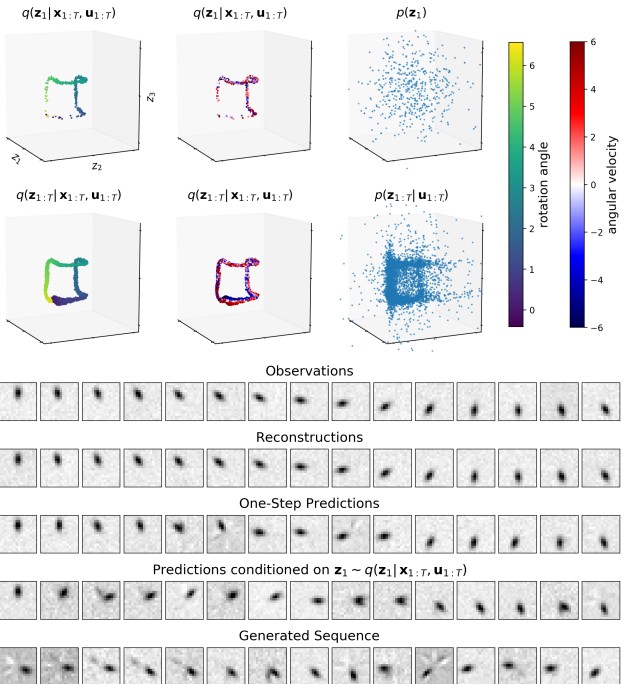

Figure 19: DKS (annealing) trained on pendulum image data (supplementary to Tab. 1a in Sec. 6.2). Without the constrained optimisation framework, DKS does not learn to accurately predict the observed dynamical system.

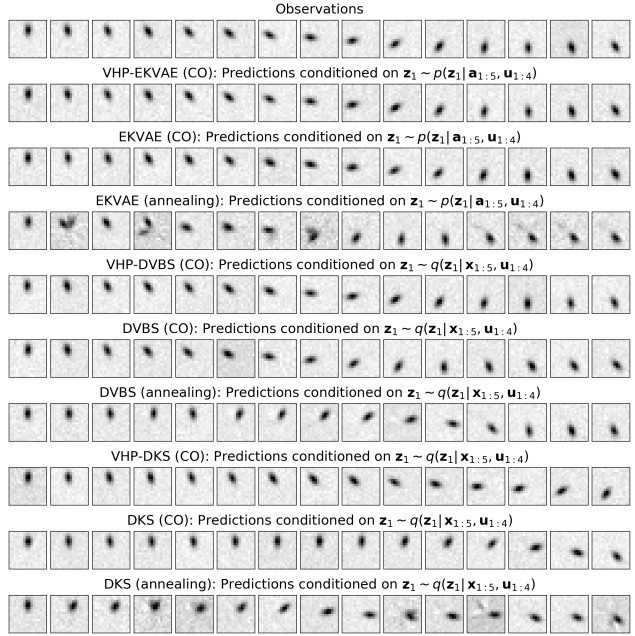

Figure 20: Summary of all models. Predicted sequences of a moving pendulum conditioned on $\mathbf{z}_1 \sim q(\mathbf{z}_1 | \mathbf{x}_{1:5}, \mathbf{u}_{1:4})$ or, in case of the EKVAE, on $\mathbf{z}_1 \sim p(\mathbf{z}_1 | \mathbf{a}_{1:5}, \mathbf{u}_{1:4})$, where the auxiliary variables are obtained through $\mathbf{a}_{1:5} \sim q(\mathbf{a}_{1:5} | \mathbf{x}_{1:5})$. The average prediction accuracy, measured by the MSE, can be found in Tab. 1a (Sec. 6.2).

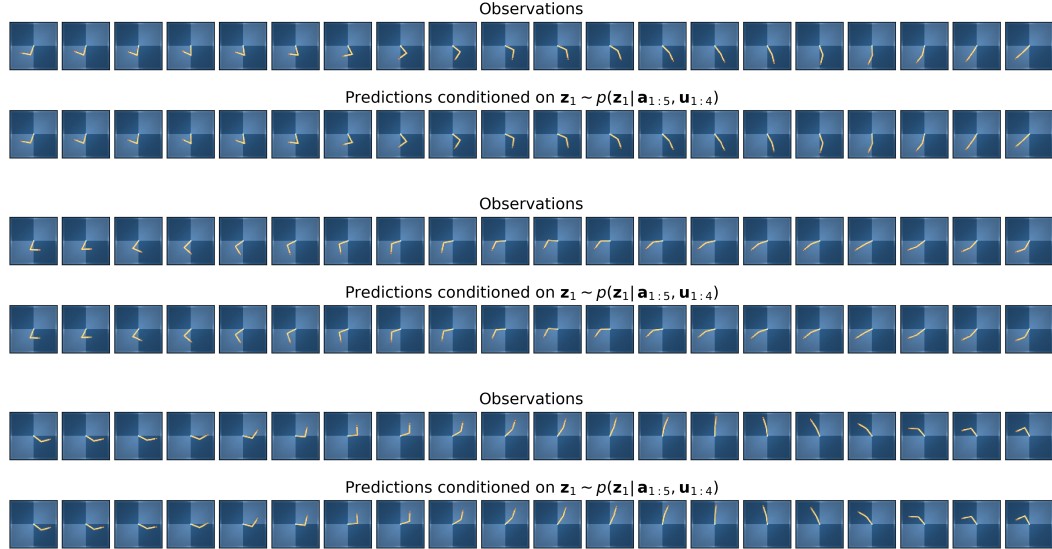

Figure 21: VHP-EKVAE (CO). Predicted sequence of a moving reacher conditioned on the smoothed distribution $\mathbf{z}_1 \sim p(\mathbf{z}_1 | \mathbf{a}_{1:5}, \mathbf{u}_{1:4})$, where $\mathbf{a}_{1:5} \sim q(\mathbf{a}_{1:5} | \mathbf{x}_{1:5})$. The average prediction accuracy is depicted in Tab. 1c (Sec. 6.2).

### A.7.3 Limitations of RNN-Based Transition Models

As a consequence of the RNN-based transition model, the KVAE [8] and the RSSM [9] learn a non-Markovian state space, i.e. not all information about the system's state is encoded in $\mathbf{z}_t$, but partially in the RNN. This is indicated in Tab. 2 (see Sec. 6.3) by the low correlation ($R^2$) between the inferred and ground-truth angular velocity, when trained on pendulum image data. The OLS regressions are performed identically to Tab. 1a (see App. A.7.2).

In order to verify that the KVAE and the RSSM do not encode the angular velocity of the pendulum in $\mathbf{z}_t$, we compare in Tab. 2 (see Sec. 6.3) the accuracy (MSE) of 500 predicted sequences $\mathbf{x}_{1:15}$ (15 time steps). In case of the KVAE, for example, these are either conditioned on the smoothed $\left\{\mathbf{z}_1^{(n)} \sim p(\mathbf{z}_1 \mid \mathbf{a}_{1:5}^{(n)}, \mathbf{u}_{1:4}^{(n)}), \ \mathbf{h}_1^{(n)}\right\}_{n=1}^{500}$ or the filtered $\left\{\mathbf{z}_5^{(n)} \sim p(\mathbf{z}_5 \mid \mathbf{a}_{1:5}^{(n)}, \mathbf{u}_{1:4}^{(n)}), \ \mathbf{h}_5^{(n)}\right\}_{n=1}^{500}$, denoted by $\mathrm{MSE}_{\mathrm{predict}}^{\mathrm{(smoothed)}}$ and $\mathrm{MSE}_{\mathrm{predict}}^{\mathrm{(filtered)}}$. This allows us to isolate the influence of the RNN on the model's prediction accuracy, as we show in Fig. 22 and Fig. 23.

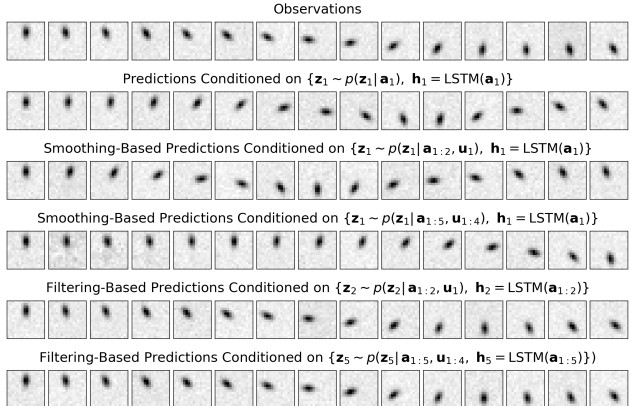

Figure 22: VHP-KVAE (CO). The predictions demonstrate that the KVAE encodes the angular velocity of the pendulum in $\mathbf{h}_t = \mathrm{LSTM}(\mathbf{a}_{1:t})$ and not in $\mathbf{z}_t$. This causes the poor smoothing-based predictions, as $\mathbf{h}_1 = \mathrm{LSTM}(\mathbf{a}_1)$ does not have access to sequence data and therefore cannot infer the angular velocity. See Tab. 2 in Sec. 6.3.

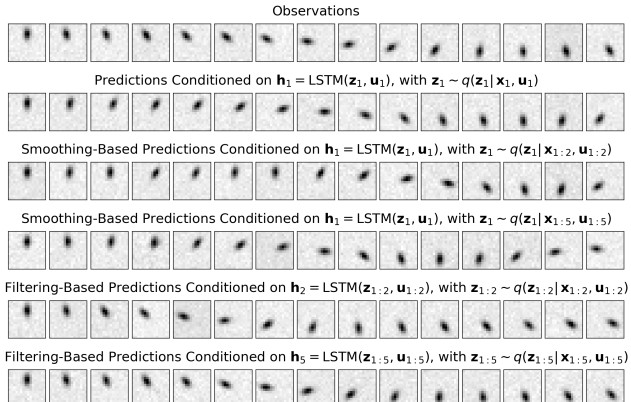

Figure 23: VHP-RSSM (CO). The predictions demonstrate that the RSSM encodes the angular velocity of the pendulum in $\mathbf{h}_t = \mathrm{LSTM}(\mathbf{z}_{1:t}, \mathbf{u}_{1:t})$ and not in $\mathbf{z}_t$. This causes the poor smoothing-based predictions, as $\mathbf{h}_1 = \mathrm{LSTM}(\mathbf{z}_1, \mathbf{u}_1)$ does not have access to sequence data and therefore cannot infer the angular velocity. See Tab. 2 in Sec. 6.3.

The KVAE uses the transition model $p(\mathbf{z}_{t+1} | \mathbf{z}_t, \mathbf{h}_t, \mathbf{u}_t)$, where $\mathbf{h}_t = \text{LSTM}(\mathbf{a}_{1:t})$; the RSSM uses the transition model $p(\mathbf{z}_{t+1} | \mathbf{h}_t)$, where $\mathbf{h}_t = \text{LSTM}(\mathbf{z}_{1:t}, \mathbf{u}_{1:t})$. Fig. 22 and Fig. 23 show that the predicted position of the pendulum in the initial time step is always identical to the observed position. Thus, we can conclude that the low accuracies of the smoothing-based predictions are due to missing information about the dynamics, i.e. the angular velocity of the pendulum. When smoothing back to the initial time step, this information can only be provided by $\mathbf{z}_1$ since the LSTM does not have access to sequential data and therefore cannot infer any dynamics. Consequently, we state that the angular velocity is encoded in $\mathbf{h}_t$ of the LSTM and can only be inferred for $t \geq 2$, as shown in Fig. 22 and Fig. 23; and verified by the different accuracies of the smoothing- and filtering-based predictions in Tab. 2 (see Sec. 6.3).

### A.7.4 Encoding Rewards: Policy Learning With Disentangled State-Space Representations

In Sec. 6.4, Fig. 6 shows the visualisation of different policies that are learned based on the disentangled (position–velocity) state-space representation of the pendulum (image data) in Fig. 5 (left). The policies are tested on the original pendulum environment that was also used to generate the dataset. The first example (top) demonstrates the pendulum swing-up, which is achieved by encoding the goal position for $r_t^{\text{pos}}(\mathbf{z}_t, \mathbf{p}_g)$ and using an action interval of $7 \geq \mathbf{a} \geq -7$. The second and third example (middle and bottom) demonstrate steady clockwise and counter-clockwise rotations of the pendulum with different angular velocities, using an action interval of $30 \geq \mathbf{a} \geq -30$. This is achieved by encoding the goal angular velocity for $r_t^{\text{vel}}(\mathbf{z}_t, \mathbf{v}_g)$: in Fig. 6 (middle), we use 50% of the maximum speed defined by the dataset; and in Fig. 6 (bottom) 85% of the maximum speed defined by the dataset. The experiments verify that the EKVAE has learned an accurate model of the pendulum. Furthermore, they demonstrate the variety of applications for disentangled (position–velocity) state-space representations and the related policy learning approach.

Fig. 24 shows visualisations of policies that are learned based on the disentangled (position–velocity) state-space representation of reacher (image data) in Fig. 3. To this end, the goal position, denoted by the red dot, was encoded to use $r_t^{\text{pos}}(\mathbf{z}_t, \mathbf{p}_g)$ (cf. Sec. 6.4). The policies are tested using the Deepmind-control-suite reacher environment. Our results show that the EKVAE has learned an accurate model of the reacher environment that avoids self-collisions and ensures precise reaching of a desired position.

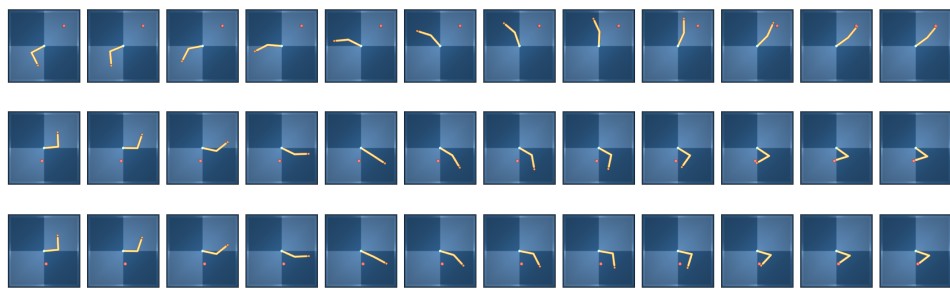

Figure 24: VHP-EKVAE (CO). Visualisation of the policy learned based on the disentangled (position–velocity) state-space representation in Fig. 3. For this purpose, the goal position (red dot) is encoded in the latent space. The results show that the EKVAE has learned an accurate model of the observed system. (see Sec. 6.4)

## A.8 Model Architectures

Table 3: Model architectures of EKVAE. FC refers to fully-connected layers.

| Dataset | Optimiser | Implementation Details | |
|---|---|---|---|
| Pendulum | Adam
$1e$-3 | Observations | 256 (flattened $16 \times 16$) |
| | | Time Steps | 15 |
| | | Actions | 1 |
| | | Auxiliary Variables | 2 |
| | | Latents | 3 |
| | | $q_\phi(\mathbf{a}_t | \mathbf{x}_t)$ | FC 128, 128, 128. ReLU activation. |
| | | $p_\theta(\mathbf{x}_t | \mathbf{a}_t)$ | FC 128, 128, 128. ReLU activation. Gaussian. |
| | | Number of Base Matrices $M$ | 16 |
| | | $\alpha$-Network | FC 64. ReLU activation. |
| | | $q_{\phi_0}(\boldsymbol{\zeta} | \mathbf{z}_1)$ | FC 64, 64. ReLU activation. |
| | | $p_{\psi_0}(\mathbf{z}_1 | \boldsymbol{\zeta})$ | FC 64, 64. ReLU activation. |
| | | Others | $\tau_1 = 10$, $\tau_2 = 0.01$, $\nu = 300$ |
| | | Batch Size | 500 |
| Reacher (angle data) | Adam
$1e$-3 | Observations | 2 |
| | | Time Steps | 30 |
| | | Actions | 2 |
| | | Auxiliary Variables | 2 |
| | | Latents | 4 |
| | | $q_\phi(\mathbf{a}_t | \mathbf{x}_t)$ | FC 128. ReLU activation. |
| | | $p_\theta(\mathbf{x}_t | \mathbf{a}_t)$ | FC 128. ReLU activation. Gaussian. |
| | | Number of Base Matrices $M$ | 8 |
| | | $\alpha$-Network | FC 64, 64. ReLU activation. |
| | | $q_{\phi_0}(\boldsymbol{\zeta} | \mathbf{z}_1)$ | FC 64, 64. ReLU activation. |
| | | $p_{\psi_0}(\mathbf{z}_1 | \boldsymbol{\zeta})$ | FC 64, 64. ReLU activation. |
| | | Others | $\tau_1 = 1$, $\tau_2 = 0.001$, $\nu = 10$ |
| | | Batch Size | 128 |
| Reacher (image data) | Adam
$5e$-3 | Observations | $64 \times 64 \times 3$ |
| | | Time Steps | 30 |
| | | Actions | 2 |
| | | Auxiliary Variables | 3 |
| | | Latents | 5 |
| | | $q_\phi(\mathbf{a}_t | \mathbf{x}_t)$ | Conv $32 \times 5 \times 5$ (stride 2), $64 \times 5 \times 5$ (stride 2),
$128 \times 5 \times 5$ (stride 2). FC 256. ReLU activation. |
| | | $p_\theta(\mathbf{x}_t | \mathbf{a}_t)$ | Deconv reverse of encoder. ReLU activation. Gaussian. |
| | | Number of Base Matrices $M$ | 8 |
| | | $\alpha$-Network | FC 64, 64. ReLU activation. |
| | | $q_{\phi_0}(\boldsymbol{\zeta} | \mathbf{z}_1)$ | FC 64, 64. ReLU activation. |
| | | $p_{\psi_0}(\mathbf{z}_1 | \boldsymbol{\zeta})$ | FC 64, 64. ReLU activation. |
| | | Others | $\tau_1 = 10$, $\tau_2 = 0.01$, $\nu = 30$ |
| | | Batch Size | 64 |

Table 4: Model architectures of DKS. FC refers to fully-connected layers.

| Dataset | Optimiser | Implementation Details | |
| --- | --- | --- | --- |
| Pendulum | Adam 1e-3 | Observations | 256 (flattened 16×16) |
| | | Time Steps | 15 |
| | | Actions | 1 |
| | | Latents | 3 |
| | | $q_\phi(\mathbf{z}_t|\mathbf{x}_{1:T}, \mathbf{u}_{1:T})$ | BiLSTM 128. sigmoid activation. FC 64. ReLU activation. |
| | | $p_\theta(\mathbf{x}_t|\mathbf{z}_t)$ | FC 128, 128, 128. ReLU activation. Gaussian. |
| | | $p_\theta(\mathbf{z}_t|\mathbf{z}_{t-1}, \mathbf{u}_{t-1})$ | FC 128, 128, 128. ReLU activation. |
| | | $q_{\phi_0}(\boldsymbol{\zeta}|\mathbf{z}_1)$ | FC 64, 64. ReLU activation. |
| | | $p_{\psi_0}(\mathbf{z}_1|\boldsymbol{\zeta})$ | FC 64, 64. ReLU activation. |
| | | Others | $\tau_1 = 10, \tau_2 = 0.01, \nu = 300$ |
| | | Batch Size | 500 |
| Reacher (angle data) | Adam 1e-3 | Observations | 2 |
| | | Time Steps | 30 |
| | | Actions | 2 |
| | | Latents | 4 |
| | | $q_\phi(\mathbf{z}_t|\mathbf{x}_{1:T}, \mathbf{u}_{1:T})$ | BiLSTM 128. sigmoid activation. FC 64. ReLU activation. |
| | | $p_\theta(\mathbf{x}_t|\mathbf{z}_t)$ | FC 128. ReLU activation. Gaussian. |
| | | $p_\theta(\mathbf{z}_t|\mathbf{z}_{t-1}, \mathbf{u}_{t-1})$ | FC 128, 128, 128. ReLU activation. |
| | | $q_{\phi_0}(\boldsymbol{\zeta}|\mathbf{z}_1)$ | FC 64, 64. ReLU activation. |
| | | $p_{\psi_0}(\mathbf{z}_1|\boldsymbol{\zeta})$ | FC 64, 64. ReLU activation. |
| | | Others | $\tau_1 = 1, \tau_2 = 0.001, \nu = 10$ |
| | | Batch Size | 128 |

Table 5: Model architectures of DVBS. FC refers to fully-connected layers.

| Dataset | Optimiser | Implementation Details | |
| --- | --- | --- | --- |
| Pendulum | Adam 1e-3 | Observations | 256 (flattened 16×16) |
| | | Time Steps | 15 |
| | | Actions | 1 |
| | | Latents | 3 |
| | | $q_\phi(\mathbf{z}_t|\mathbf{x}_{t:T}, \mathbf{u}_{t:T})$ | LSTM 128. sigmoid activation. FC 64. ReLU activation. |
| | | $p_\theta(\mathbf{x}_t|\mathbf{z}_t)$ | FC 128, 128, 128. ReLU activation. Gaussian. |
| | | Number of Base Matrices $M$ | 16 |
| | | $\alpha$-Network | FC 64. ReLU activation. |
| | | $q_{\phi_0}(\boldsymbol{\zeta}|\mathbf{z}_1)$ | FC 64, 64. ReLU activation. |
| | | $p_{\psi_0}(\mathbf{z}_1|\boldsymbol{\zeta})$ | FC 64, 64. ReLU activation. |
| | | Others | $\tau_1 = 10, \tau_2 = 0.01, \nu = 300$ |
| | | Batch Size | 500 |
| Reacher (angle data) | Adam 1e-3 | Observations | 2 |
| | | Time Steps | 30 |
| | | Actions | 2 |
| | | Latents | 4 |
| | | $q_\phi(\mathbf{z}_t|\mathbf{x}_{t:T}, \mathbf{u}_{t:T})$ | LSTM 128. sigmoid activation. FC 64. ReLU activation. |
| | | $p_\theta(\mathbf{x}_t|\mathbf{z}_t)$ | FC 128. ReLU activation. Gaussian. |
| | | Number of Base Matrices $M$ | 8 |
| | | $\alpha$-Network | FC 64, 64. ReLU activation. |
| | | $q_{\phi_0}(\boldsymbol{\zeta}|\mathbf{z}_1)$ | FC 64, 64. ReLU activation. |
| | | $p_{\psi_0}(\mathbf{z}_1|\boldsymbol{\zeta})$ | FC 64, 64. ReLU activation. |
| | | Others | $\tau_1 = 1, \tau_2 = 0.001, \nu = 10$ |
| | | Batch Size | 128 |

Table 6: Model architectures of KVAE. FC refers to fully-connected layers.

| Dataset | Optimiser | Implementation Details | |
|---|---|---|---|
| Pendulum | Adam 1e-3 | Observations | 256 (flattened 16×16) |
| | | Time Steps | 15 |
| | | Actions | 1 |
| | | Auxiliary Variables | 2 |
| | | Latents | 3 |
| | | $q_\phi(\mathbf{a}_t|\mathbf{x}_t)$ | FC 128, 128, 128. ReLU activation. |
| | | $p_\theta(\mathbf{x}_t|\mathbf{a}_t)$ | FC 128, 128, 128. ReLU activation. Gaussian. |
| | | Number of Base Matrices $M$ | 16 |
| | | $\alpha$-Network | FC 64. ReLU activation. |
| | | Dynamics Parameter Network | LSTM 64. sigmoid activation. |
| | | $q_{\phi_0}(\boldsymbol{\zeta}|\mathbf{z}_1)$ | FC 64, 64. ReLU activation. |
| | | $p_{\psi_0}(\mathbf{z}_1|\boldsymbol{\zeta})$ | FC 64, 64. ReLU activation. |
| | | Others | $\tau_1 = 10, \tau_2 = 0.01, \nu = 300$ |
| | | Batch Size | 500 |

Table 7: Model architectures of RSSM. FC refers to fully-connected layers.

| Dataset | Optimiser | Implementation Details | |
|---|---|---|---|
| Pendulum | Adam 1e-3 | Observations | 256 (flattened 16×16) |
| | | Time Steps | 15 |
| | | Actions | 1 |
| | | Latents | 3 |
| | | $q_\phi(\mathbf{z}_t|h_{t-1}, \mathbf{x}_{t:T}, \mathbf{u}_{t:T})$ | LSTM 128. sigmoid activation. FC 64. ReLU activation. |
| | | $p_\theta(\mathbf{x}_t|\mathbf{z}_t)$ | FC 128, 128, 128. ReLU activation. Gaussian. |
| | | $p_\theta(\mathbf{z}_t|h_{t-1})$ | FC 128, 128, 128. ReLU activation. |
| | | Deterministic State Model | LSTM 64. sigmoid activation. |
| | | $q_{\phi_0}(\boldsymbol{\zeta}|\mathbf{z}_1)$ | FC 64, 64. ReLU activation. |
| | | $p_{\psi_0}(\mathbf{z}_1|\boldsymbol{\zeta})$ | FC 64, 64. ReLU activation. |
| | | Others | $\tau_1 = 10, \tau_2 = 0.01, \nu = 300$ |
| | | Batch Size | 500 |