# OpenReview forum: "Latent Matters: Learning Deep State-Space Models"
_NeurIPS.cc/2021/Conference — NeurIPS 2021 Poster_

### Official Review · Reviewer_Qj5e · 2021-07-08

**Rating:** 6
**Confidence:** 3

**Summary:**

This paper has two main contributions.
1. The authors adapt REWO for helping with disentanglement of learned representations and improving predictive accuracy
2. The authors replace the RNN encoding of the nonlinearity in [6] by the locally linear model of [11-12] while keeping closed form inference

The authors then illustrate how combining these together leads to better accuracy in learned latent dynamics for a set of models, in particular they show that their approach can recover the unobserved velocity as a latent variable in motion models.

**Limitations And Societal Impact:**

The authors do not study the impact of the hyperparameters on the learned models, the most critical one being in my opinion $\mathcal{D}_0$, as per the authors description, is critical in learning predictive model.

I believe the authors methodology only applies to model with unimodal latent posteriors (due to their linearisation format)

The authors only consider somewhat short time series (30 time steps at most). It is not clear how this gradient based learning would work with longer datasets.


**Main Review:**

Overall I believe this paper shows promising empirical results, however it should be carefully revisited in order to be publishable.

Section 3

The statement "Thus, one can assume that initial-state samples uniformly cover the manifold of the learned latent representation" l97-98 is not clear. How is this assumption/statement a consequence of batching/splitting time series?
I would argue that Algo 1 should be reproduced in this section, as it stands, you list it as one of your main contributions, but the details are absent from the core of the paper.


Section 4

The authors mention that the main reason they resort to learning a surrogate Jacobian matrix is essentially computational complexity as using AD to compute it would require $D_z$ backward passes. I agree that this is indeed computationally expensive. However, as it stands their proposed parametrisation is not related to extended linearisation but rather to sigma-points one, which, similarly to the authors suggestion considers a linearisation $F$ as a weighted sum of "base" matrices. The main difference between the two approaches being that while the authors consider the weights to depend on $z,u$ while sigma-point methods implicitely consider that the "base" matrices $F^{(m)}$ do instead.

This consideration makes the claim that $F_{\psi} \approx J_f$ feel wrong to me and makes the name EKVAE a bit misleading. If anything the authors are trying to jointly learn the best "basis" of transition matrices $F^{(m)}$ at the same time as they try to learn how to project $f$ on it. This is indeed a valid idea, but would in my opinion deserve to be compared with standard sigma-points techniques applied to an explicit transition model. I expect these would be highly competitive for at least three reasons: sigma-points methods are highly parallelisable on GPU and don't require expensive Jacobian calculations, the models considered are fairly low-dimensional both in $z$ and $a$ (as per Appendix 8 if I understand it correctly), and the model posterior distribution should be unimodal by design.

In Eq 15, the authors mention that they use MC to integrate $z_{t-1}$ out in order to compute the observation log-likelihood increment. However this contradicts the fact that they are assuming that $z_t, z_{t-1}$ are jointly Gaussian, so that they should be able to marginalise it in closed form too?

The authors mention that they chose to use a fixed matrix $H = \delta_{ij}$ instead of learning it as was done in e.g. [6]. I agree that this is useful for disentanglement as it reduces unidentifiability within the model. It is however unclear if the resulting improvement of EKVAE over KVAE shown in section 6 comes from this only or from better encoding of the state transitions (except if the authors chose to run KVAE experiments with H (C in [6]) fixed too?).

Section 6

The most concerning aspect of this section consists in the choice of the hyperparameter $\mathcal{D}_0$, for which the authors give no principled rule or heuristics. Was it optimised for or curated in some way? If so this would make the comparison with fixed annealing schedules if these were not curated. Additionally the impact of changing $\mathcal{D}_0$ should be studied at least on the simplest model so as to provide the reader with a clear understanding of performance improvement/deterioration.

The phrasing of what the experiments exactly are is a bit unclear (e.g. l276-278), this would be explained better by wrtting it more formally: what exactly are the authors sampling from? Additionally I don't understand why the authors are conditioning on the latent $z_1$ to measure predictive accuracy when conditioning on the observations up to time $t$ in order to predict the future would make more sense. Moreover, the way I understand this is that the conditioned $z_1$ was fixed once and for all and not sampled from, which makes the values in the tables a, b, c random. If I am correctly understanding it makes it hard to know if the improvement is statistically significant.


Comments below did not affect my evaluation but would need to be corrected in the event of acceptance.

Typos:
l18: "sequence data" should probably be "sequential data"
l90:  It is common practise
l116: In the initial phase (upper case)
l119: the VHP $(\psi_0, \phi_0)$  and the transition model $\phi_0)$  ---> these are the *parameters* of the models you want to optimise
l189: range of application; result in a poorly trained

There are a certain number of issues with the bibliography, a non-exhaustive list being:
- Kalman and Tung are missing first names
- [5] is broken
- several middle names are missing the full stop, e.g. [15], [18]

## Update:
After having read other reviews and authors responses (in particular the back and forth on my comments), I have improved my rating from 4 to 6. I still believe there are a number of issues with the presentation of the paper which would warrant a more thorough correction than the rebuttal format allows. However, I now believe that this work could be published given the minor additional corrections suggested, although would be more beneficial to the research community if it was partially reorganised (I have comments similar to that of reviewer SDjn).

**Time Spent Reviewing:**

13

---

> ### Author Response · Authors · 2021-08-10
> **Response 1**
>
> We would like to thank the reviewer for the invested time and for the valuable feedback. In the following, we will address each question separately:
>
> *“The statement "Thus, one can assume that initial-state samples uniformly cover the manifold of the learned latent representation" l97-98 is not clear. How is this assumption/statement a consequence of batching/splitting time series?”*
>
> **R1**: Batching multiple time series treats these sequences as iid. Splitting/extracting sub-sequences at (uniformly) random time points leads to the implicit assumption that every state marginal is identical, because the same initial prior is used at different time points. The same can be said for each conditional distribution $p(z_t | z_t-1, u_t-1)$.
> A good example for explaining this is the pendulum dataset: classically an observed time series of the pendulum would start at rotation_angle=0 and angular_velocity=0. However, in order to improve gradient-based optimisation in DSSMs, time series with a length of, say, 60 are divided into 4 series of length 15. This is a valid approach in Markovian models since, to put it simply, we want to learn the transition $p(z_t|z_t-1, u_t-1)$ (cf. R14). As a consequence of the splitting, the initial states of the last three time series can now have any possible rotation angle and angular velocity. Therefore, if the dataset is large enough, initial-state samples uniformly cover the manifold of the learned latent representation. This can be seen, for example, in Fig 11 (top row, first two visualisations).
>
> *"I would argue that Algo 1 should be reproduced in this section, as it stands, you list it as one of your main contributions, but the details are absent from the core of the paper."*
>
> **R2**: We agree with your suggestion and will add Alg. 1 to the core paper if the page limit is increased for the revised version.
>
> *"The authors mention that the main reason they resort to learning a surrogate Jacobian matrix is essentially computational complexity as using AD to compute it would require $D_z$ backward passes. I agree that this is indeed computationally expensive. However, as it stands their proposed parametrisation is not related to extended linearisation but rather to sigma-points one, which, similarly to the authors suggestion considers a linearisation $F$ as a weighted sum of "base" matrices. The main difference between the two approaches being that while the authors consider the weights to depend on z,u while sigma-point methods implicitely consider that the "base" matrices $F^{(m)}$ do instead.
> This consideration makes the claim that $F_{\psi} \approx J_f$ feel wrong to me and makes the name EKVAE a bit misleading. ..."*
>
> **R3**: There is indeed a close relationship to sigma-points linearisation, which is the subject of our current research. However, our goal is to improve on/compare to the DVBS [11] and the KVAE [6]. Both models are based on the transition-model approach we implemented in the EKVAE. The difference is that DVBS does not take advantage of closed-form Bayesian inference, leading to a less accurate dynamic model, as we verify in Tab. 1. The KVAE on the other hand uses a linear Gaussian transition w.r.t. $z_t$, which requires an additional RNN for modelling nonlinear dynamics and leads to a non-Markovian state-space, where Kalman smoothing does not work well (cf. Tab. 2). We clearly state in L133--137 (and in L151--152) that we do not perform a Taylor-expansion based linearisation in the EKVAE, but use the locally-linear transition introduced in [26] to implicitly learn F. Moreover, we derive in App. A.5 that this allows us to apply the extended Kalman filter/smoother algorithm but avoid a computationally expensive linearisation (first-order Taylor expansion) in the prediction step, which motivates the term EKVAE. At the same time, we agree that this name might be misleading as the reader could expect a Taylor-expansion based linearisation, and we are open to rename the EKVAE.\
> Furthermore, we would like to emphasise that the claim in L149 (“$F_{\psi} \approx J_f$”) was introduced in [26] (right after Eq. (4)), which is a widely cited work in DSSM literature.\
> Regarding the very interesting idea of using the mentioned sigma-point method (unscented Kalman filter/smoother): we would be happy to investigate this in future work.
>
> *"In Eq 15, the authors mention that they use MC to integrate $z_{t−1}$ out in order to compute the observation log-likelihood increment. However this contradicts the fact that they are assuming that $z_t,z_{t−1}$ are jointly Gaussian, so that they should be able to marginalise it in closed form too?"*
>
> **R4**: This is an important detail of the EKVAE. It is possible to compute $p(a_t|a_{1:t − 1}, u_{1:t − 1})$ analytically, but not favourable. In our initial experiments, we marginalised $z_{t-1}$ also in closed form; however, this led to a poorly trained transition model. We found that this is because $p(z_t|z_{t-1}, u_{t-1})$ is only optimised via extended Kalman smoothing, i.e. via deterministic mean values, when both $z_t$ and $z_{t-1}$ are marginalised in closed form (see L186--189). This does not cover the area of application since we use the probabilistic samples of the transition function when predicting an observed system. For this reason, we found a way to decouple $z_t$ from $z_{t-1}$ (see L195--200) allowing (realised through MC) a sample-based optimisation of the transition model.
>
> *"The authors mention that they chose to use a fixed matrix $H = \delta_{ij}$ instead of learning it as was done in e.g. [6]. I agree that this is useful for disentanglement as it reduces unidentifiability within the model. It is however unclear if the resulting improvement of EKVAE over KVAE shown in section 6 comes from this only or from better encoding of the state transitions (except if the authors chose to run KVAE experiments with H (C in [6]) fixed too?)."*
>
> **R5**: We did not use $H = \delta_{ij}$, but a fully learnable H for the pendulum experiments in Tab. 1a. We will emphasise this in the paper and point out that the disentanglement is an additional/useful option that leads to better interpretability and allows encoding the reward function---but it is not a requirement for our method. We applied the disentanglement ($H = \delta_{ij}$) only for the image reacher experiments in Tab. 1c (which allows visualising the 5D latent space) and for the model-based reinforcement learning experiments in Sec. 6.4. In general, we have not seen in our experiments a difference in the prediction accuracy of EKVAE when disentangling the latent space. Moreover, the KVAE experiments in Sec. 6.3 show that the EKVAE indeed learns a better encoding of the state/identifies the observed system (compare $R^2$ values in Tab. 1a and Tab. 2).
>
> *"The most concerning aspect of this section consists in the choice of the hyperparameter $D_0$, for which the authors give no principled rule or heuristics. Was it optimised for or curated in some way? If so this would make the comparison with fixed annealing schedules if these were not curated."*
>
> **R6**: The heuristic for finding $D_0$ is fairly simple: we first determined the best distortion $D_\text{max}$ that the respective model can achieve when trained via the original/classical approach; then we defined $D_0 = 0.9 D_\text{max}$. We have not spent a lot of time in hyperparameter search since the experiments worked well right away. In our experiments, the CO framework was not sensitive w.r.t. the hyperparameters. Moreover, we used the same hyperparameters across all models (without the need of a hyperparameter search). This is supported by the evaluations in Fig. 9, where additionally different annealing schedules are compared to CO.
>
> *"Additionally the impact of changing $D_0$ should be studied at least on the simplest model so as to provide the reader with a clear understanding of performance improvement/deterioration."*
>
> **R7**: We will add an extensive analysis of how $D_0$ affects the quality of learned models.
>
> *"The phrasing of what the experiments exactly are is a bit unclear (e.g. l276-278), this would be explained better by wrtting it more formally: what exactly are the authors sampling from?"*
>
> **R8**: For each of the 500 sequences in the test set, we compute an individual $q(z_1|x_{1:5}, u_{1:4})$ (or in case of the EKVAE $p(z_1|a_{1:5}, u_{1:4})$, where $a_{1:5} \sim q(a_{1:5}|x_{1:5})$ is sampled once for each sequence in the test set). Then, we sample an individual $z_1$ for each sequence; note that the sampled $z_1$ contain the information of the first 5 observations because we use smoothing distributions. These samples---visualisations can be found, for example, in Fig. 11--19 (top row, first to visualisations)---are then used as conditions to predict the observed sequences. The MSE values in the last column in Tab. 1 and 2 measure how close the predicted sequences are to the observed ones. We will describe this process more clearly in the paper.

---

> > ### Author Response · Authors · 2021-08-10
> > **Response 2**
> >
> > *"Additionally I don't understand why the authors are conditioning on the latent $z_1$ to measure predictive accuracy when conditioning on the observations up to time $t$ in order to predict the future would make more sense."*
> >
> > **R9**: We use smoothers, and conditioning on the smoothing distribution $q(z_1|x_{1:5}, u_{1:4})$ provides more information than conditioning on the filtered distribution $q(z_5|x_{1:5}, u_{1:4})$ (the latter is usually used when the model is just a filter). This is because conditioning on $q(z_1|x_{1:5}, u_{1:4})$ additionally tells us whether the smoothing worked/the model could infer an accurate state-space representation of partially observed systems already in the initial time step. This is, for example, important in the KVAE experiments in Sec. 6.3: Fig. 3 shows that conditioning on the smoothing distribution leads to significantly less accurate predictions than conditioning on the filtered distribution (which is because the angular velocity is encoded in the LSTM, see L592--601 for a completion of this argument).
> >
> > *"Moreover, the way I understand this is that the conditioned $z_1$ was fixed once and for all and not sampled from, which makes the values in the tables a, b, c random. If I am correctly understanding it makes it hard to know if the improvement is statistically significant."*
> >
> > **R10**: We have partly addressed this question in **R8**. Since we sample an individual $z_1$ for each of the 500 sequences in the test set, the applied metrics are meaningful (and also used in previous publications, e.g. http://proceedings.mlr.press/v97/becker-ehmck19a/becker-ehmck19a.pdf, Tab. 1). Therefore, the improvements are statistically significant. We hope this clarifies the question and would be happy to discuss this point in more detail if required.
> >
> > *"Typos: ... ... bibliography..."*
> >
> > **R11**: We highly appreciate that the reviewer pointed out the typos and double-checked the bibliography. We would like to thank him once again for his time investment. The mentioned issues are resolved.
> >
> > *"The authors do not study the impact of the hyperparameters on the learned models, the most critical one being in my opinion $D_0$ as per the authors description, is critical in learning predictive model."*
> >
> > **R12**: In our experiments, the value of $D_0$ was not critical for learning a good predictive model (cf. **R6**). It is critical though to formulate the sequential ELBO as the Lagrangian of a CO problem. We will be happy to add an extensive analysis of how $D_0$ affects the quality of learned models.
> >
> > *"I believe the authors methodology only applies to model with unimodal latent posteriors (due to their linearisation format)"*
> >
> > **R13**: Indeed, we rely on the assumption that the underlying dynamics of the observed system are deterministic. This is the case in, for example, all environments in the DeepMind Control Suite [24], and all models/papers we compare to or discuss in related work [e.g. 2, 6, 7, 11, 12, 15, 16, 26] are based on this assumption; thus, require only a unimodal posterior distribution. However, this is a very important point the reviewer mentioned and especially interesting in several forecasting scenarios where we cannot assume that it is possible to learn the complete underlying dynamics or they do not exist. Another example is SLAM. Therefore the application of multimodal posterior distributions is subject of our current research.
> >
> > *"The authors only consider somewhat short time series (30 time steps at most). It is not clear how this gradient based learning would work with longer datasets."*
> >
> > **R14**: This is an important point concerning most DSSMs and all models we compare to [e.g. 6, 7, 11, 12, 15]. As addressed in L95--L97 and in **R1**, gradient-based learning is one of the main reasons for splitting (long) observed time series into short sequences. This is a valid approach for Markovian models since we are interested in learning $p(z_t|z_{t-1}, u_{t-1})$. The only limitation that might occur with short sequences is that they are not long enough to allow the model to infer the true underlying state $z_t $of the observed system (e.g. for inferring the angular velocity of the pendulum, access to at least two observations is necessary). However, we show in our experiments that the underlying state has been inferred/identified (cf. $R^2$ values in Tab. 1).
> >
> > We hope that we have been able to dispel some doubts and would appreciate it if the reviewer would reconsider the rating. At the same time, we will be happy to answer any further questions.

---

> > ### Comment · Reviewer_Qj5e · 2021-08-25
> > **Specific response to R3**
> >
> > Thank you for the detailed answer, here I will focus on the specific R3 point (and in particular the claim that the parametric linearisation matrices learn the Jacobian).
> > There is a subtle but important difference in your methodology and the methodology you referred to ([26]). The section 2.2 in [26] considers a
> > problem where the form of the transition model is known. Under this assumption, iteratively computing the linearisation as the Jacobian of the function at the previous estimate of the best linearisation trajectory $\bar{z}_t$ converges to a global optimum for some loss function. In a non-control setting it can be understood perhaps more easily via https://ieeexplore.ieee.org/abstract/document/250476.
> > The claim in your paper is somewhat different. It says that the estimated globally optimal linearisation matrices will converge to the Jacobians of the ssm (abstract) functions estimated at the state of the previous time step. I don't believe there is any reason for this to be true, but if it is, this claim would require at least empirical evidence on a toy model.

---

> > > ### Author Response · Authors · 2021-08-26
> > > **Response to the specific response to R3**
> > >
> > > We would like to thank the reviewer once again for the time investment and hope that we can conclude from the reviewer's response that the remaining questions (R1, R2, R4--R14) have been satisfactorily addressed.
> > >
> > > W.r.t. R3: we use a "locally-linear" transition function [11, 26]; thus, F has to represent a local linearisation of the observed non-linear dynamical system in order to achieve meaningful predictions (per definition). Our claim is that we can learn this transition function via closed-form Bayesian inference, namely extended Kalman filtering/smoothing. To apply extended Kalman filtering/smoothing, the local Jacobian is required. In our derivation (A.5), we replace the local Jacobian by $F_\psi$ (cf. Eq. (47--52)) to compute the filtered and smoothed distributions. Thus, if $F_\psi$ would not approximate the local Jacobian, our EKF/EKS-based approach would not lead to learning a precise transition function. In terms of empirical evidence: the EKVAE learns a very precise transition function (e.g. Fig. 10 for pendulum image data and Fig. 20 for reacher RGB image data). Averaged over the whole testset, it outperforms the compared models by a factor of 3 to 10 in terms of prediction accuracy (see Tab. 1).

---

> > > > ### Comment · Reviewer_Qj5e · 2021-09-03
> > > > **RE**
> > > >
> > > > I am pretty certain that the globally optimal (for standard losses like MSE) linearisations for the filtering or smoothing problems are not the Jacobian evaluated at the previous filtering mean. It has been proven in numerous papers including the one I linked above. While I understand that you are metaphorically doing something "similar" to using the Jacobian, saying that your scheme approximates the Jacobian is wrong. I could however get behind a statement along the lines of "our scheme tries to globally find the best linearisation at each time step for the task at hand" (please don't use my exact words but make it fit with your text).
> > > >
> > > > If you disagree, you can verify using a numerical example along the lines of generating toy data from using a sine as a transition matrix (or something similar), then use your scheme on this data, and show that the learned 1D matrix is exactly cosine evaluated at the previous filtering mean. However I just don't see how this can be true.
> > > >
> > > > EDIT: Let me be more positive in my assessment. I believe you are learning a better linear representation of the function than the (locally optimal) Taylor expansion at the previous state mean.

---

> > > > > ### Author Response · Authors · 2021-09-05
> > > > > **RE**
> > > > >
> > > > > We agree with this line of argument and will modify the relevant passages to bring them in line with the proposed statement.

---

> > ### Author Response · Authors · 2021-09-03
> > **Update R7 and R12**
> >
> > We have added an extensive analysis to the paper of how $D_0$ affects the quality of learned models. As a result, we have achieved an equivalent performance---in terms of $R^2$ values and predication accuracy---when using a deviation of up to $\pm10$% from the original $D_0$ value. The CO approach is therefore not very sensitive with regard to $D_0$.

---

> > > ### Comment · Reviewer_Qj5e · 2021-09-03
> > > **RE**
> > >
> > > Thanks! That's good to know. Have you also added this small explanation too (or similar)? My opinion is that this is very important to tell the reader, otherwise finding D_0 in the first place is just a shot in the dark, no matter if results are sensitive to its value or not..
> > >
> > > "The heuristic for finding D_0 is fairly simple: we first determined the best distortion D_max that the respective model can achieve when trained via the original/classical approach, etc"

---

> > > > ### Author Response · Authors · 2021-09-03
> > > > **RE**
> > > >
> > > > Yes, the heuristic is now described in the experimental part of the core paper. For the experiment mentioned, we refer the reader to the appendix.

---

> > ### Comment · Reviewer_Qj5e · 2021-09-03
> > **Re R1**
> >
> > This is more clear now. as I understand it you are essentially trying to learn the model in a "stationary" regime. It would however be good to maybe rephrase the sentence slightly. As it stands the statement "Thus, one can assume that initial-state samples uniformly cover the manifold of the learned latent representation" sounds like a model assumption, while it is more of a methodological assumption. I would essentially like to see the "Thus" be detailed. Can you simply go for "In order to [your explanation], one can assume that initial-state samples uniformly cover the manifold of the learned latent representation"?

---

> > > ### Author Response · Authors · 2021-09-05
> > > **RE**
> > >
> > > This is a good point, we will rephrase the sentence as suggested.

---

### Official Review · Reviewer_SDjn · 2021-07-09

**Rating:** 6
**Confidence:** 4

**Summary:**

The authors make three contributions. 1) They apply the constrained optimization framework for VAE type models to the deep state space model case. 2) They add in the hierarchical prior approach to model the initial distribution for the first state. 3) They propose a model with linear transition matrices given by a neural network weighting learned basis matrices. Auxiliary observation variables are also introduced to allow for a linear observation model along with an extra encoder/decoder. This means filtering and smoothing distributions can be computed analytically.

**Limitations And Societal Impact:**

The limitations of the specific contributions could be explained more thoroughly e.g. dependence on hyperparameters, linearity assumptions etc.

**Main Review:**

Overall, I think this paper has some interesting ideas, however, when reading the paper it is hard to distil the paper to a contribution that can be easily built upon and used by other researchers. The paper feels like it should be split into two papers meaning each contribution in isolation can be more thoroughly explained and evaluated.

Beginning with the constrained optimization framework, judging by the experiments included, this looks to be very helpful for learning consistent and interpretable transition functions. In experiments it looks to consistently and significantly improve performance, not only in the author’s proposed model, but in all the baselines too. I think this is a very interesting observation and feel like it should be explored further. For example, are there any heuristics for setting the desired reconstruction quality D_0  parameter and what effects does this hyperparameter have on the quality of learned models. Also, it may be interesting to look at how the rate-distortion framework from http://proceedings.mlr.press/v80/alemi18a/alemi18a.pdf can be interpreted from this sequential learning viewpoint.

Along with this, the authors also propose a new model class - the EKVAE - which I feel could be a paper in its own right. The CO framework can be integrated with a wide variety of models so I am unsure about why these two contributions are in the same paper, it makes the overall message hard to discern. It would be good to have some more intuition on the design choices for the EKVAE for example why it was chosen that each matrix should be a weighting of a set of base matrices, did this give better performance than other models in preliminary experiments?

Finally the hierarchical prior does seem to help models learn a better initialization for their generative model. Though in some cases, e.g. Fig. 8 it seems the model can still learn a useful transition model even with the naïve prior. This then brings into question whether the added complexity of the hierarchical prior is useful. I think this should be taken into account in the experiments as it is important to separate out the benefit of the prior and the transition model. For short sequences e.g. pendulum then the marginal better results with the prior could be just due to the first two frames as opposed to a higher quality transition model. For the longer reacher dataset if it was just the first two frames then the marginal difference between having the prior and without would be even less but these results are not included in the table. Overall, I think it would be good to discuss whether the prior just helps the first 1 or 2 states or actually results in a higher quality transition model.

Regarding clarity, I think the paper is well written and organized. Each of the claims are demonstrated in their own experiment though I feel each claim could do with more expansive investigation (see above comments).

**Edit after rebuttal**

I have read the author responses and other reviews. I believe the paper has good ideas and encourage the authors to continue working on this problem but I believe the paper is not quite ready for publication yet in its current format. This is due to the ideas being presented in a way that makes it more difficult for other researchers to be build on them.  I think the paper would greatly benefit from a clearer focus on single elements of the proposed design or by achieving improvements on a significant problem that could not be achieved if any one of the proposed elements was not included.

**2nd update**
After further consideration of author responses and other reviews, I have increased my score to 6. I believe the ideas presented are novel and with a bit extra motivation of how the proposed methods in the paper are related and both necessary in a camera ready, I think it warrants publishing.


**Time Spent Reviewing:**

8

---

> ### Author Response · Authors · 2021-08-10
> **Response**
>
> We would like to thank the reviewer for the invested time and for the valuable feedback. In the following, we will address each question separately:
>
> *"Overall, I think this paper has some interesting ideas, however, when reading the paper it is hard to distil the paper to a contribution that can be easily built upon and used by other researchers. The paper feels like it should be split into two papers meaning each contribution in isolation can be more thoroughly explained and evaluated."*
>
> **R1**: We have considered splitting the paper into two papers. However, our goal is to show how to learn DSSMs in order to achieve high prediction accuracies, and how the prediction accuracy correlates with the learned state-space representation. In our experiments, we demonstrate that both the CO framework and the EKVAE improve the quality of the learned state-space representation (see $R^2$ values in Tab. 1) leading to significantly higher prediction accuracies (cf. Tab. 1, last column).
>
> *"Beginning with the constrained optimization framework, judging by the experiments included, this looks to be very helpful for learning consistent and interpretable transition functions. In experiments it looks to consistently and significantly improve performance, not only in the author’s proposed model, but in all the baselines too. I think this is a very interesting observation and feel like it should be explored further. For example, are there any heuristics for setting the desired reconstruction quality D_0 parameter and what effects does this hyperparameter have on the quality of learned models."*
>
> **R2**: Thank you very much! In our experiments, the CO framework was not sensitive w.r.t. the hyperparameters, which is also supported by the evaluation in Fig. 9 (right). Moreover, we used the same hyperparameters across all models (without the need for a hyperparameter search).
> The heuristic for finding $D_0$ is fairly simple: we first determined the best distortion $D_\text{max}$ which the respective model can achieve when trained via the original/classical approach; then we defined $D_0 = 0.9 D_\text{max}$. As already mentioned, we have not spent a lot of time in hyperparameter search since the experiments worked well right away. However, we would be happy to add an extensive analysis of how $D_0$ affects the quality of learned models.
>
> *"Also, it may be interesting to look at how the rate-distortion framework from http://proceedings.mlr.press/v80/alemi18a/alemi18a.pdf can be interpreted from this sequential learning viewpoint."*
>
> **R3**: There is indeed a close connection, which we partly emphasise in Sec. 2 (L62). The Lagrangian defined in Eq. (4) and the constrained optimisation problem in Eq. (5) can be interpreted as the “sequential” versions of the rate-distortion framework from [1] / http://proceedings.mlr.press/v80/alemi18a/alemi18a.pdf (cf. Section 4, 1st paragraph in [1]). However, we motivate the connection between the ELBO and the Lagrangian in more detail by means of the EM algorithm following the line of argument in [14]. We will add a description in Sec. 3 highlighting the connection to the framework in [1] and pointing out that our CO framework extends the theory introduced in [1] to sequential models. Thank you very much for the useful hint.
>
> *"Along with this, the authors also propose a new model class - the EKVAE - which I feel could be a paper in its own right. The CO framework can be integrated with a wide variety of models so I am unsure about why these two contributions are in the same paper, it makes the overall message hard to discern. It would be good to have some more intuition on the design choices for the EKVAE..."*
>
> **R4**: While the CO framework is a standalone approach, the EKVAE performs significantly worse without the CO framework (cf. Fig. 13, predictions and generations). This is due to the design choices we made for the EKVAE allowing it to highly benefit from the CO framework: since the state-space is optimised based on samples from the auxiliary-variable space, it is important to first learn a meaningful encoding in the auxiliary-variable space. This is ensured by the constraint-based optimisation of the Lagrangian since the distortion is defined by the encoder-decoder pair that infers the auxiliary variables.
>
> *"... for example why it was chosen that each matrix should be a weighting of a set of base matrices, did this give better performance than other models in preliminary experiments?"*
>
> **R5**: The transition model we use was already introduced in [26] in the context of DSSMs. From our experience it is easier to train than a vanilla MLP, especially if “matrices” are required as in the case of the Kalman filter/smoother algorithm. This is also one of the reasons why several other works rely on this transition model (e.g. [6] and [16]). Moreover, our goal was to improve on/compare to the DVBS [11] and the KVAE [6]. Both models are based on this transition model approach; however, DVBS does not take advantage of closed-form Bayesian inference, leading to a less accurate dynamic model, as we verify in Tab. 1. The KVAE on the other hand uses a linear Gaussian transition w.r.t. $z_t$, which requires an additional RNN for modelling nonlinear dynamics. We show in Sec. 6.3 that this not only reduces the prediction accuracy of the model, but also leads to a non-Markovian state-space where the model does not encode all information in $z_t$. With the EKVAE, we address these issues.
>
> *"Finally the hierarchical prior does seem to help models learn a better initialization for their generative model. Though in some cases, e.g. Fig. 8 it seems the model can still learn a useful transition model even with the naïve prior. This then brings into question whether the added complexity of the hierarchical prior is useful. I think this should be taken into account in the experiments as it is important to separate out the benefit of the prior and the transition model. For short sequences e.g. pendulum then the marginal better results with the prior could be just due to the first two frames as opposed to a higher quality transition model. For the longer reacher dataset if it was just the first two frames then the marginal difference between having the prior and without would be even less but these results are not included in the table. Overall, I think it would be good to discuss whether the prior just helps the first 1 or 2 states or actually results in a higher quality transition model."*
>
> **R6**: In Tab. 1 a, b, c (last column), we demonstrate that the prediction accuracy significantly increases when the VHP (more complex prior) is applied. Fig. 10 shows that the difference in the prediction accuracy is due to a more precise transition model when trained with the VHP. This is because of less over-regularisation during training that affects the whole transition model. Thus, the VHP does not only improve the first 1--2 states, but the quality of the transition model.
> Fig. 8 demonstrates that a predefined prior can furthermore lead to a broken generative model, where the transition model is not optimised to process samples from the prior. As a result, unrealistic sequences are generated.
> In our view, a complex prior is therefore a very important modification, and to the best of our knowledge, there is no other published work in the DSSM literature demonstrating these effects on the quality of the learned model.

---

> > ### Author Response · Authors · 2021-09-03
> > **Update R2**
> >
> > We have added an extensive analysis to the paper of how $D_0$ affects the quality of learned models. As a result, we have achieved an equivalent performance---in terms of $R^2$ values and predication accuracy---when using a deviation of up to $\pm10$% from the original $D_0$ value. The CO approach is therefore not very sensitive with regard to $D_0$.

---

> ### Author Response · Authors · 2021-09-06
> **Response to the edited review**
>
> We appreciate the reviewer's feedback.
>
> "*achieving improvements on a significant problem that could not be achieved if any one of the proposed elements was not included*"
>
> To the best of our knowledge, however, identifying the system dynamics of the reacher environment---which requires a 5D latent space (see Fig. 2)---has not yet been shown, especially not on the basis of high-dimensional RGB image data. This was only possible by combining the EKVAE with the CO-framework and led to very accurate and realistic predictions (cf. Tab. 1c and Fig. 20). Other models have not been able to achieve this, even in combination with the CO-framework, as we pointed out in **R17** (response to reviewer xnvk).

---

### Official Review · Reviewer_xnvk · 2021-07-12

**Rating:** 7
**Confidence:** 4

**Summary:**

This paper proposes various improvements to deep sequential latent variable models (or deep state space models). Namely, 1) the constrained optimization method from the Taming VAEs paper [Rezende & Viola, 2018], 2) a method for improving empirical priors over the first time step’s latent variable (referred to as VHP), and 3) a new model that combines the locally linear mappings of Karl et al., 2017 and the KVAE variable separation from Fraccaro et al., 2017. These improvements are demonstrated on pendulum and reacher datasets (both with images and joint angles), showing improvements in system identification, prediction accuracy, and the ability to use the state representations from MBRL. Overall, the paper provides several helpful tips for constructing and training deep state space models.

**Limitations And Societal Impact:**

The limitations of the paper are not addressed in-depth. The authors claim that they have shown that “prediction accuracy strongly depends on the model itself.” I would like to see some discussion around how other models could be used for model-based RL and whether this technique could be scaled to more complex settings, such as natural video or audio.

There do not appear to be specific negative societal impacts associated with this paper.

**Main Review:**

**Originality**: The paper centers on applying several ideas developed for static latent variable models to the dynamic setting. Likewise, the proposed model (extended Kalman VAE) draws heavily from Fraccaro et al.’s KVAE and locally linear mappings that have been used previously for sequential latent variable models, e.g., Karl et al. 2017. The main novelty of this paper is in demonstrating that the aforementioned techniques are also useful in learning sequential models. While I believe that these findings are useful, they are not particularly surprising, as the previous techniques are not specific to static models. One point of originality that I appreciated was the application of this technique to the reacher environment, both with pixel observations and for the purposes of model-based control. This is helpful for connecting the sequential generative modeling community with the RL community.


**Quality**: The paper is of fairly high quality, combining multiple complex techniques and demonstrating improvements over several baseline models. Implementing and evaluating these techniques is non-trivial and at the forefront of current research. The experiments in the paper also help to illuminate some overlooked aspects of sequential latent variable models, particularly tied to representation learning, e.g., the importance of constrained optimization and the Markovian latent spaces. The experiments analyze multiple metrics across two dynamics environments.


**Clarity**: The writing is overall quite clear. My main concern here is regarding the complexity of the resulting model and inference/learning procedure. The paper combines multiple complex ideas: constrained optimization, variational hyperprior (VHP), locally linear dynamics, and the already fairly complex Kalman VAE. Given the number of elements in this paper, the main message of the paper is rather dispersed. Including diagrams of the various techniques and model/inference dependencies would be immensely helpful for parsing the paper, even if these were included in the appendix. Additionally, some of the main points of the paper, e.g., the benefit of constrained optimization for representation learning, are only apparent from results tables. The figures in the main paper currently only demonstrate the ability to learn disentangled latent representations. Figure 7 from the appendix, which compares the latent representations between constrained optimization and annealing, strikes me as a stronger demonstration; I would consider moving this to the main paper.


**Significance**: The significance of the paper is one of my larger concerns. Again, given the number of components to the final model, it is somewhat difficult to assess where the benefits are coming from and the take-aways for the generative modeling community. While some of the qualitative and quantitative results are clear, e.g., the benefits of constrained optimization for system identification, the significance of other quantitative results, e.g., around VHP, are less obvious. For instance, how meaningful is a difference of 1 (assuming nat) in test ELBO? The authors also present results using their model for model-based RL. Yet, it is unclear how this differs in capabilities from other models, i.e., any generative model could, in principle, be used for model-based planning. The authors rely on particular assumptions around encoding reward functions in latent space and conditioning their parametric policy on the latent variable. Thus, while I appreciate the model-based RL experiments, this doesn’t strike me as a significant contribution.

**Other Points**:

- In equation 1, the Markov assumption is not necessarily standard for modern models. I would consider a more general formulation in which this assumption is dropped.
- Line 59: the recognition model shouldn’t necessarily need to learn the dynamics. E.g., Marino et al. present an amortized filtering method that does not require learning the dynamics within the inference model.
- Eq. 2: The variety of script F’s and L’s is somewhat confusing.
- Eq. 5: Previous methods (as well as this formulation) don’t actually use EM. Instead, they use amortized variational inference, amortizing the E-step optimization in an inference model.
- Line 91: It would be useful to define what “over-regularization” means here.
- Line 102: The acronym VHP is not defined (variational hyper-prior).
- Line 103: Why just the initial time step?
- Line 106: When will this bound be tight?
- Line 116: Is this two-stage procedure well-justified? It seems somewhat ad-hoc.
- Section 4.3 was difficult to follow.
- Section 5: There are some previous works in sequential VAEs that are not cited, e.g., Chung et al., 2015.
- Line 221: these are not necessarily the papers I would cite for stochastic gradient variational Bayes, as they make the additional assumption of amortizing inference. Other papers, such as BBVI (Ranganath et al.), may be more appropriate.
- Figure 1: Wasn’t lambda supposed to be in [0,1]?
- Table 1c: Why aren’t there other models here?
- Figure 3: I like this example. but isn’t it a bit unfair to compare while conditioning on 1 time step? In practice, such models would be used after a “burn in” of multiple time steps.
- Section 6.4: the benefits of this particular model for model-based RL could be explained better. It’s unclear from the writing. Also, the goal encoding process is unclear.

Marino et al., 2018. A general method for amortizing variational filtering.;
Chung et al., 2015. A recurrent latent variable model for sequential data.;
Ranganath et al., 2013. Black box variational inference.

---
**Update**: I have read the other reviews and the authors' response. Each of the other reviewers raised separate concerns regarding the paper. Reviewer oVeY raised concerns regarding the motivation of the ideas in the paper. I agree with the authors, though, regarding the justification of the Markov assumption and constrained optimization. Reviewer SDjn felt that the paper was somewhat disjoint in its presentation. The (lack of) cohesion of the paper, in my view, is not such a concern as to prevent readers from understanding the concepts and take-home points. Finally, reviewer Qj5e raised various concerns regarding the model/evaluation. The authors provided a detailed response to these questions, none of which struck me as serious issues with the paper. Thus, I do not see the points raised by the other reviewers as meaningfully detracting from the paper, and I am keeping my score at 7. This paper explores several useful ideas for dynamics modeling with deep latent variable models, which are demonstrated with a thorough empirical evaluation. The research community would benefit from the ideas in this paper, thus, in my view, warranting publication.

**Time Spent Reviewing:**

3.5

---

> ### Author Response · Authors · 2021-08-10
> **Response 1**
>
> We would like to thank the reviewer for the invested time and for the valuable feedback. In the following, we will address each question separately:
>
> *“Including diagrams of the various techniques and model/inference dependencies would be immensely helpful for parsing the paper, even if these were included in the appendix.”*
>
> **R1**: This is a good suggestion. We will add diagrams that help understanding our model to the appendix.
>
> *“Additionally, some of the main points of the paper, e.g., the benefit of constrained optimization for representation learning, are only apparent from results tables. The figures in the main paper currently only demonstrate the ability to learn disentangled latent representations. Figure 7 from the appendix, which compares the latent representations between constrained optimization and annealing, strikes me as a stronger demonstration; I would consider moving this to the main paper.”*
>
> **R2**: We agree that Fig. 7 is a good visualisation of the second and third column in Tab. 1a., and it would be great if we had the space to add it to the main paper. For the submission, we had decided that Fig. 1 is more important as it shows the optimisation process when using the CO framework, i.e. the importance of first learning a good reconstruction/encoding the rotation angle, which then forms the basis for inferring the angular velocity. Fig. 2 shows the capability of the EKVAE when combined with the CO framework (without CO we do not learn a good model, cf. Tab. 1c). To the best of our knowledge, system identification on the reacher RGB image data has not been shown yet. Note that the disentanglement in Fig. 2 is only necessary in order to visualise the learned 5D latent space.
>
> *“... how meaningful is a difference of 1 (assuming nat) in test ELBO?”*
>
> **R3**: As we emphasise in the abstract (L4--5) and the introduction (L25--26), the ELBO is a suboptimal metric for determining whether the model has learned to predict the observed system (cf. Tab. 1). This was our main motivation for formulating the ELBO as the Lagrangian of a CO problem. Thus, the most important metric that shows the benefits of the CO framework is the prediction accuracy of the learned models (last column in the tables) measured by the MSE.
>
> *“The authors also present results using their model for model-based RL. Yet, it is unclear how this differs in capabilities from other models, i.e., any generative model could, in principle, be used for model-based planning.”*
>
> **R4**: In order to use a model for RL, it has to have a high prediction accuracy. Otherwise a policy learned on the basis of this model will not work when applied to the original/observed environment. The EKVAE achieves the highest prediction accuracy, which makes it a better choice compared to the other models. In Fig. 5 and Fig. 6, we demonstrate the high precision of the EKVAE by testing the learned policies on the original environments.
>
> *“The authors rely on particular assumptions around encoding reward functions in latent space and conditioning their parametric policy on the latent variable. Thus, while I appreciate the model-based RL experiments, this doesn’t strike me as a significant contribution.”*
>
> **R5**: We see the possibility of encoding rewards (which is interesting if we do not have access to rewards) as a side contribution. This is only possible if the latent space is disentangled, as described in Sec. 6.4.
>
> *“In equation 1, the Markov assumption is not necessarily standard for modern models. I would consider a more general formulation in which this assumption is dropped.”*
>
> **R6**: The Markov assumptions lead to state-space models. State-space models, while not currently in favor in the deep learning community, are still a go-to model in the engineering disciplines. The widespread use of Kalman filters and their variations is proof of that. This is why we believe that EKVAEs are relevant, and why we focus on state-space models in our analysis. This choice is also reflected by our experimental setups.
>
> *“Line 59: the recognition model shouldn’t necessarily need to learn the dynamics. E.g., Marino et al. present an amortized filtering method that does not require learning the dynamics within the inference model.”*
>
> **R7**: By “learning the dynamics”, we did not mean to say that the recognition model needs to learn $p(z_{t+1}|z_t, u_t)$ explicitly. However, a good posterior, like the true posterior, needs to be aware of the dynamics. If this is not achieved by explicit weight sharing, the recognition model will implicitly have to learn the same concepts and dynamics as the prior. This is true for most of the related literature, including Marino et al. A notable exception is Karl et al. ([11]), who first noticed this particular challenge for sequential variants of VAEs.
>
> *“Eq. 5: Previous methods (as well as this formulation) don’t actually use EM. Instead, they use amortized variational inference, amortizing the E-step optimization in an inference model.”*
>
> **R8**: In classical constrained optimisation, the constraint D(θ, φ) is not optimised. To bridge the gap to amortised variational inference, the authors in [14] motivated the CO approach by variational EM, where minimising the optimisation objective can be viewed as the M-step and minimising D(θ, φ) is part of the E-step. Although in amortised VI the recognition and generative model are trained jointly, the above perspective justifies the usage of CO in the context of amortised VI.
>
> *“Line 91: It would be useful to define what “over-regularization” means here.”*
>
> **R9**: We will describe over-regularisation in more detail. In case of a predefined $p(z_1)$, the KL term in the ELBO regularises $q(z_1|x_{1:T}, u_{1:T})$ towards $p(z_1)$. In case of, for example, a standard normal distribution and pendulum data, we either do not obtain the barrel shape (over-regularisation) or the generative model is broken: the transition model is not optimised to process samples from the prior, as described in Section 3.1 (cf. figures in App. A.7.1: $q(z_1|x_{1:T}, u_{1:T})$ vs $p(z_1)$).
>
> *“Line 102: The acronym VHP is not defined (variational hyper-prior).”*
>
> **R10**: We will add the definition of the acronym: Variational Hierarchical Prior (VHP).
>
> *“Line 103: Why just the initial time step?”*
>
> **R11**: $q(\zeta|z_1)$ and $p(z_1|\zeta)$ allow modelling an arbitrary complex $p(z_1)$ by using samples form $q(z_1|x_{1:T}, u_{1:T})$. As a consequence, $q(z_1|x_{1:T}, u_{1:T})$ is neither over-regularised nor the generative model is broken (cf. figures in App. A.7.1).
> From another perspective: $p(z_1)$ sets the overall shape of the manifold, for which Gaussians are not sufficiently complex. $p(z_t|z_{t-1}, u_{t-1})$ models then only local changes, for which Gaussians are usually appropriate.
>
> *“Line 106: When will this bound be tight?”*
>
> **R12**: The bound is tight if the bound in Eq. (6) is tight (see Eq. (21--22) for details).
>
> *“Line 116: Is this two-stage procedure well-justified? It seems somewhat ad-hoc.”*
>
> **R13**: Yes it is justified---a detailed explanation can be found in [14] (after Eq. 10).
>
> *“Section 5: There are some previous works in sequential VAEs that are not cited, e.g., Chung et al., 2015.”*
>
> **R14**: We agree and we will add a discussion to Sec. 5.
>
> *“Line 221: these are not necessarily the papers I would cite for stochastic gradient variational Bayes, as they make the additional assumption of amortizing inference. Other papers, such as BBVI (Ranganath et al.), may be more appropriate.”*
>
> **R15**: To the best of our knowledge, the term ‘stochastic gradient variational Bayes’ was introduced by Kingma et al. in [13]. As we apply the method introduced in [13, 22] (which differs from BBVI) to DSSMs, we decided to highlight [13] and [22].
>
> *“Figure 1: Wasn’t lambda supposed to be in [0,1]?”*
>
> **R16**: That’s an important point: in order to optimise a lower bound on the sequential ELBO, lambda is supposed to be in [0,1]. Therefore, REWO ensures that lambda is in [0,1] at the end of training (cf. L113 and App. A2). In the beginning of the training it is important to have a high lambda to optimise the reconstruction quality/satisfy the constraint.
>
> *“Table 1c: Why aren’t there other models here?”*
>
> **R17**: The RGB image data requires the use of CNNs (without CNNs, we were not able to learn models capable of accurate predictions). The recognition model of the EKVAE is just an MLP, which can easily be replaced by a CNN. By contrast, the recognition models of DKS and DVBS are based on RNNs. In our experiments, we have not been able to achieve good results neither by combining the RNN-based recognition models with CNNs nor by using convolutional RNNs. Since CNNs are not part of the original implementations in [15] and [11], we decided to not add these results to the table and to highlight that the EKVAE is capable of learning an accurate prediction based on high-dimensional RGB image data.
>
> *“Figure 3: I like this example. but isn’t it a bit unfair to compare while conditioning on 1 time step? In practice, such models would be used after a “burn in” of multiple time steps.”*
>
> **R18**: Thank you! We use smoothing posteriors; thus, $z_1 \sim p(z_1| a_{1:5}, ...)$ should contain the same information as $z_5 \sim p(z_5| a_{1:5}, ...)$. However, Tab. 2 (see prediction accuracy measured by the MSE) and Fig. 3 show that this is not the case, which can only be explained if the angular velocity is encoded in the LSTM and not in $z_t$. A completion of this argument can be found in App. A.7.3 (L592--L601). The goal of the experiments in Sec. 6.3 is to highlight this limitation---because depending on what the learned latent representation/model is used for, this can have a large impact on the outcome.

---

> > ### Author Response · Authors · 2021-08-10
> > **Response 2**
> >
> > *“Section 6.4: the benefits of this particular model for model-based RL could be explained better. It’s unclear from the writing. Also, the goal encoding process is unclear.”*
> >
> > **R19**: The EKVAE achieves the highest prediction accuracies of all compared models. Additionally it is capable of disentangling the latent space, which allows learning policies without having access to rewards (just by encoding a goal state). We will improve the description of the encoding process, thank you very much for this feedback.

---

### Official Review · Reviewer_oVeY · 2021-07-16

**Rating:** 7
**Confidence:** 3

**Summary:**

This paper proposes a framework for learning neural network-based state-space models from raw data. In comparison to the standard RNN-based approach [1], there are several modifications: (i) a modified ELBO formulated in the context of a constrained optimization (CO) framework [2]. CO prioritizes reconstruction term at the beginning of the training; (ii) hierarchical prior [3] instead of a simple Gaussian; (iii) a modified Kalman-VAE framework [4] instead of RNN-based transition model. Experimental evidence on pendulum data and the reacher environment indicates that the resulting framework outperforms baselines in predictive modeling and ELBO values.


[1] A Recurrent Latent Variable Model for Sequential Data

[2] Taming VAEs

[3] Learning Hierarchical Priors in VAEs

[4]  A Disentangled Recognition and Nonlinear Dynamics Model for Unsupervised Learning

**Ethical Concerns:**

None.

**Limitations And Societal Impact:**

The authors have addressed the limitations. There are no obvious concerns in terms of societal impact.

**Main Review:**

*Originality*: The resulting framework is a combination of three existing ideas, as indicated in the summary above. Additional modifications were introduced to adapt the CO framework to the sequential setting and to replace LSTMs in the Kalman-VAE model in the spirit of [5]. Overall, the originality is somewhat limited. This is generally not a big issue in my mind, as long as combining the ideas from the previous work is well-justified.

*Motivation:* My main issue with this paper is motivation. The authors claim that each introduced modification is beneficial for improving the predictive model of the described dynamical system. However, I found it difficult to understand why would introduced changes make a substantial difference. For example, the authors claim that 'RNN-based transition models as in [6, 7] can lead to a non-Markovian state-space, where the latent variables do not capture the entire information about the system’s state.'. However, I do not find this well-justified and it is completely unclear to me why this would be the case. The authors also claim that 'As a consequence of the RNN-based transition models, the angular velocity is not encoded in zt but in the RNN." which is also unclear. It is also unclear to me why CO is necessary -- why it would be beneficial to optimize the reconstruction term (the distortion) first and then the KL term (the rate). Using a more complex prior can certainly improve ELBO, but this is a very minor modification that leads to non-surprising results.
I would be willing to improve my score if the authors convinced me why the proposed modifications are necessary and beneficial.
At the present state, the overall justification of the proposed framework boils down to experimental results.

*Clarity:* The paper is fairly well-written overall and relatively easy to follow. It is also well-structured.

*Quality:* The methodology seems technically sound.  Combined with the supplementary material, there are several interesting experimental studies. Including more baselines would make the work more convincing.

*Significance:* The general topic is relevant. The presented results are performed on the synthetic data so it is difficult to judge the practical significance. One big drawback is that the source code is missing, especially since the overall framework is fairly complex and hence difficult to implement from scratch.

[5] Deep Variational Bayes Filters: Unsupervised Learning of State Space Models from Raw Data

**Update:** Raised the score to 7.

**Time Spent Reviewing:**

5

---

> ### Author Response · Authors · 2021-08-10
> **Response**
>
> We would like to thank the reviewer for the invested time and for the valuable feedback. In the following, we will address each question separately:
>
> *“My main issue with this paper is motivation. The authors claim that each introduced modification is beneficial for improving the predictive model of the described dynamical system. However, I found it difficult to understand why would introduced changes make a substantial difference. For example, the authors claim that 'RNN-based transition models as in [6, 7] can lead to a non-Markovian state-space, where the latent variables do not capture the entire information about the system’s state.'. However, I do not find this well-justified and it is completely unclear to me why this would be the case. The authors also claim that 'As a consequence of the RNN-based transition models, the angular velocity is not encoded in zt but in the RNN." which is also unclear.”*
>
> **R1**: In the case of the pendulum, for example, the information about the rotation angle and the angular velocity are required to predict the future state. RNN-based transition functions $p(z_{t+1}|z_t, h_t, u_t)$ are conditioned on $z_t$ and $h_t$. As a consequence, the future state is predicted based on both the latent state and the RNN. Thus, in practice, the above-mentioned state information (rotation angle, angular velocity) is stored in the deterministic RNN states that are not regularised to enforce the probabilistic assumptions of the graphical model such as Markovian dynamics. For example, see Sec. 6.3 where we demonstrate that the angular velocity of the pendulum is encoded in the RNN hidden state $h_t$: since we use smoothing posteriors, $z_1 \sim p(z_1| a_{1:5}, ...)$ should contain the same information as $z_5 \sim p(z_5| a_{1:5}, ...)$. However, Tab. 2 (see prediction accuracy measured by the MSE) and Fig. 3 show that this is not the case as the predictions are much more accurate when conditioned on $z_5$. This can only be explained if the angular velocity is encoded in the RNN ($h_t$) and not in $z_t$. Hence, the latent variables do not capture the entire information about the state. A completion of this argument can be found in App. A.7.3 (L592--601).
>
>
> *“It is also unclear to me why CO is necessary -- why it would be beneficial to optimize the reconstruction term (the distortion) first and then the KL term (the rate).”*
>
> **R2**: In the case of the pendulum, for example, the model needs to first encode the rotation angle in order to infer the angular velocity. The encoding of the rotation angle is incentivised by optimising the reconstruction term/constraint. In Fig. 1, we demonstrate the optimisation process when using the CO framework: until epoch 90 the model mainly improves the encoding of the rotation angle (see colour code in the top row); the constraint is satisfied approximately at epoch 70. From epoch 110 one can see that the model inferred the angular velocity (see colour code in the bottom row)---this is realised by optimising the KL term. All models trained via CO achieve a significantly higher prediction accuracy (see last column in Tab. 1). This correlates with a better system identification (measured by the $R^2$ values in Tab. 1). In our view, the prediction accuracy of the models is the most important metric to validate the quality of the model, which clearly demonstrates the benefit of our CO approach.
>
>
> *“Using a more complex prior can certainly improve ELBO, but this is a very minor modification that leads to non-surprising results.“*
>
> **R3**: As we emphasise in the abstract (L4--5) and the introduction (L25--26), the ELBO is a suboptimal metric for determining whether the model has learned to predict the observed system. This was our main motivation for formulating the ELBO as the Lagrangian of a CO problem. In Tab. 1, we demonstrate that the prediction accuracy significantly increases when the VHP (more complex prior) is applied---this is due to less over-regularisation. Furthermore, we show in Fig. 8 that a predefined prior can lead to a broken generative model, where the transition model is not optimised to process samples from the prior. As a result, unrealistic sequences are generated. In our view, a complex prior is therefore a very important modification, and to the best of our knowledge, there is no other published work in the DSSM literature demonstrating these effects on the quality of the learned model.
>
>
> *“I would be willing to improve my score if the authors convinced me why the proposed modifications are necessary and beneficial.”*
>
> **R4**: We hope that we have been able to address the concerns and will be happy to answer any further questions.

---

> > ### Comment · Reviewer_oVeY · 2021-08-31
> > **Response**
> >
> > I would like to thank the authors for their response.
> >
> > **R1:** If I understood correctly, the authors' reply explains why learning SSMs is preferable to learning RNNs. Indeed, this has been observed in the previous works too and I agree with the authors. Thank you for the clarification.
> >
> > **R2:** From the authors' reply, I can infer that the main motivation for the CO framework is the intuition behind how the pendulum dynamics should be learned (angle first, angular velocity after), and the corresponding experimental results that support the intuition. While I tend to agree with the authors, I have to acknowledge that it is unclear whether and how this intuition translates to real-life settings.
> >
> > **R3:** I do understand that ELBO is not the appropriate metric if the aim is to obtain a predictive model that predicts well across many time steps in the future. I also understand why VHP would lead to learning better predictive models. While I would still argue that introducing a more complex prior is a relatively minor contribution, I agree that illuminating the problems with an improper choice of the prior could be beneficial for the practitioners.
> >
> > **R4:** Overall, I feel that the motivation for introducing CO and VHP to learn better DSSM is based on intuition, and is supported by the experimental results. One issue is that the experiments were conducted only on toy data. Moreover, the source code is missing and this is the main reason why I was hesitant to improve my score. Nevertheless, I read the remaining responses and will raise my score to 7.

---

> > > ### Author Response · Authors · 2021-09-06
> > > **RE**
> > >
> > > Thank you very much for the positive feedback.
> > >
> > > **R2**: This intuition can be transferred to any dynamical system where only part of the state can be inferred through a single observation---and a sequence of observations is required to infer the full state (this also applies to real-life settings). In this case, the CO framework allows first learning the part of the state that can be inferred through a single observation (e.g. rotation angle, position, etc.)---which is then used to facilitate learning/inferring the remaining part of the state (e.g. velocity, acceleration, etc.) using the whole sequence.

---

### Decision · Program_Chairs · 2021-09-27

**Decision:**

Accept (Poster)

**Comment:**

This paper develops a methodology for learning deep state-space models to obtain accurate temporal predictions of observed dynamical systems. This is achieved by combining various techniques: a constrained optimization (CO) method (along the lines of the GECO method of Rezende & Viola, 2018), a method to improve the prior for the first latent (VHP) and a modified extended Kalman-VAE (EKVAE) framework (akin to Karl et al., 2017). The authors show that the combination of these techniques provide improvements over baselines in predictive modeling and ELBO values.

After the rebuttal, two reviewers have increased their score (Reviewers oVeY and Qj5e). The final scores are 7, 7, 6, 6.
The main concerns raised by the reviewers in the discussion, in particular Qj5e and SDjn (as well as by Reviewer xnvk in their review) is that the paper lacks focus. While CO applied to state-space models is interesting, the addition of th EKVAE and VHP seems more marginal and makes the resulting model and inference/learning procedure fairly complex. The message of the paper is not quite clear and it is difficult to parse as a result. Because of the different elements, it was felt that crucial details were only mentioned in passing.  The paper would benefit from having an improved presentation (see suggestions by Reviewer xnvk) and, if the authors want to keep  CO, VHP and EKVAE in the paper, a significant application where all the proposed elements are necessary (CO, VHP and EKVAE).

This paper was discussed at length between the AC and SAC.